# *SparseSSM*: Efficient Selective Structured State Space Models Can Be Pruned in One-Shot

Kaiwen Tuo [1 2]   Huan Wang [1]

## Abstract

State-space language models such as Mamba match Transformer quality while permitting linear complexity inference, yet still comprise billions of parameters that hinder deployment. While existing one-shot pruning methods are effective for generic linear and attention blocks, they are not designed with the overall Mamba architecture in mind and fail to account for the time-shared and discretized state-transition matrix at the heart of the selective state-space module (SSM). In this paper, we introduce *SparseSSM*, the first one-shot pruning framework that extends the classic optimal brain surgeon (OBS) framework to state space architectures. Our layer-wise algorithm **(i)** derives an approximate second-order saliency score that aggregates Hessian-trace information across time steps, **(ii)** incorporates a component sensitivity analysis to guide feed-forward network (FFN) pruning, which also sheds light on where redundancy resides in mamba architecture, **(iii)** can be easily extended to semi-structured and structured sparsity, and generalized to other SSM-based architectures. Empirically, we prune 50% of SSM weights without fine-tuning and observe only limited performance degradation, achieving the current state-of-the-art one-shot pruning algorithm for Mamba-based LLMs.

## 1. Introduction

The rapid expansion of Transformer-based large language models (LLMs), which now scale to hundreds of billions of parameters (Touvron et al., 2023; Zhang et al., 2022; Workshop et al., 2022), has created an urgent demand for efficient model compression (Feng et al., 2025; Shao et al.,

2025). The deployment of such models involves substantial computational cost and environmental impact. Among various compression techniques, network pruning, the removal of redundant weights, remains a classic yet effective method to reduce model size and accelerate inference with minimal performance degradation (LeCun et al., 1989; Hassibi & Stork, 1992; Han et al., 2016; Ma et al., 2023; Frantar & Alistarh, 2023; Feng et al., 2024; Ren et al., 2026). However, many pruning approaches, especially those based on magnitude or gradient information, require retraining or fine-tuning to recover accuracy (Han et al., 2015; Li et al., 2017; He et al., 2017), which is feasible for smaller models but becomes prohibitively expensive on the scale of modern LLMs. To address this, researchers have introduced **one-shot** pruning strategies that induce sparsity in one shot without any additional optimization, in addition to more traditional **training-based** pipelines when budget allows (Lee et al., 2019; Tanaka et al., 2020; Frantar & Alistarh, 2022; Zhu et al., 2025; Su & Wang, 2026).

In the Transformer regime, lots of methods have emerged to support one-shot pruning without fine-tuning, achieving surprisingly strong performance. Notably, SparseGPT (Frantar & Alistarh, 2023) introduced an approximate optimal brain surgeon (OBS) (Hassibi & Stork, 1992) inspired framework that prunes massive LLMs to over 50% sparsity in a single pass with negligible degradation. SparseGPT leverages local second-order information to reconstruct pruned weights and minimize output error. Simpler alternatives such as Wanda (Sun et al., 2023) offer lightweight heuristics based on the product of magnitude and input activation per output neuron, yet still achieve accuracy competitive with more sophisticated OBS-based approaches. These methods exemplify a growing trend in Transformer-based LLMs compression: **OBS-guided pruning** that can operate efficiently at scale while preserving LLM quality, even at high sparsity levels (Meng et al., 2024).

Recently, Mamba (Gu & Dao, 2023; Dao & Gu, 2024) has emerged as a promising state space alternative to Transformers, replacing the attention mechanism with a selective state space model (SSM) which enables linear complexity sequence processing and significantly faster inference. Mamba-based LLMs have demonstrated competitive performance compared to Transformer-based LLMs of similar

---

[1]Westlake University, Hangzhou, China [2]Tongji University, Shanghai, China. Correspondence to: Huan Wang <wanghuan@westlake.edu.cn>.

*Proceedings of the 43rd International Conference on Machine Learning*, Seoul, South Korea. PMLR 306, 2026. Copyright 2026 by the author(s).

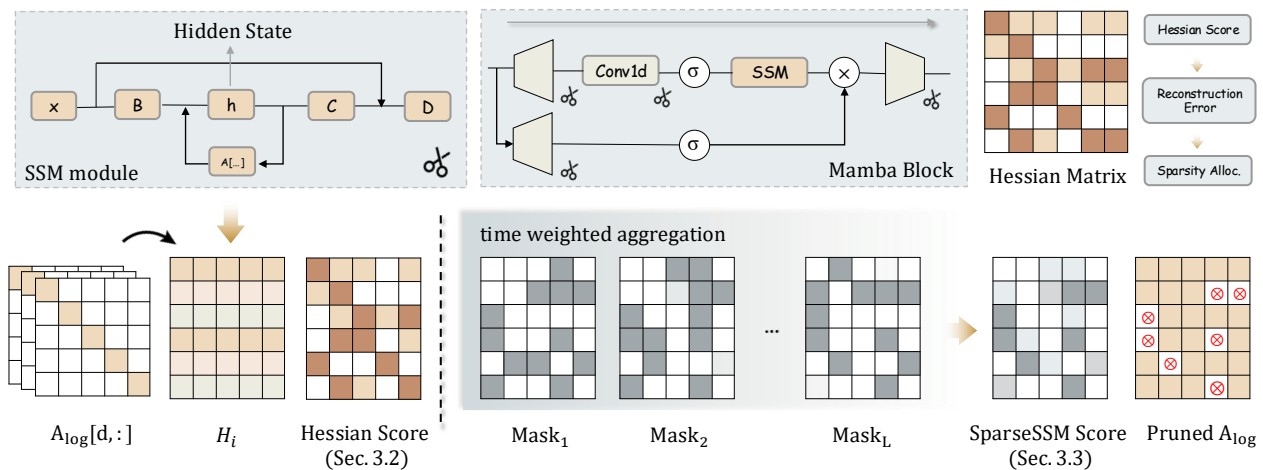

*Figure 1.* Illustration of *SparseSSM*. The **first row** illustrates the standard forward pass of a classic Mamba block. The right part highlights that we collect the Hessian matrix during the forward pass to guide the sparsity allocation across different FFN modules. In the **second row**, the **left panel** shows the procedure for obtaining a mask from the Hessian estimate at a single time step (see Section 3.2), while the **right panel** presents our weighted strategy for merging the masks across all time steps, darker background indicates larger weights (see Section 3.3).

scale (Zuo et al., 2024). Despite their efficiency and effectiveness, Mamba-based LLMs still contain billions of parameters in the FFN components and rely on time-shared parameters in the SSM module, which can incur non-trivial inference latency in certain cases. To date, most pruning research has focused on Transformer-based models, with limited efforts targeting Mamba or other state-space architectures. This motivates a key question: **Can we design a one-shot pruning method specifically tailored to the Mamba architecture?**

However, most existing pruning techniques developed for feed-forward and attention layers of Transformers cannot be directly transferred to the Mamba architecture, due to its time-shared parameters, discretization operation, and FFN modules with highly heterogeneous sensitivity to pruning. For example, **(i)** The parameter $A$ in the SSM module is time-shared, implying that any pruning decision must account for importance metrics computed at each time step; unlike spatial aggregation, however, the activations at one step are directly influenced by the previous time step. **(ii)** During execution, $A$ is discretized into $\Delta A$; therefore, pruning must explicitly consider this discretization operation. **(iii)** In the Mamba architecture, the robustness of different FFN modules to pruning varies substantially.

To address these issues, we proposed *SparseSSM*, a layerwise pruning method that generalizes the traditional OBS framework to efficient selective structured state space models (see figure 1). Our technical **contribution** can be summarized as follow:

- We introduce *SparseSSM* that adapts the classic optimal brain surgeon framework to the Mamba architecture. Our

method computes approximate second-order weight importance for the time-sharing SSM parameters, proposes a weighted mask aggregation method to address the time-sharing nature of the SSM module, and mitigates the highly heterogeneous pruning sensitivity across FFN submodules. To our best knowledge, this is the first application of OBS-based pruning to Mamba's architecture.

- We further adapt *SparseSSM* to semi-structured and structured sparsity formats, achieving substantial speedups and practical efficiency gains, and we generalize our method to other classic SSM-based architectures, including vision models and hybrid Mamba-Transformer models.

- *SparseSSM* achieves significantly superior performance compared to current state-of-the-art one-shot pruning algorithms for Mamba-based LLMs. Through experiments on language modelling benchmarks, we demonstrate that our method can prune 50% of the SSM weights and observe only limited performance degradation without finetuning.

## 2. Related Work

**Selective State Space Models.** Selective State Space Models (SSMs) have emerged as promising alternatives to the attention layers in Transformers, particularly due to their computational complexity in linear time and their ability to handle long-range dependencies efficiently (Gu & Dao, 2023; Dao & Gu, 2024). Unlike traditional attention-based mechanisms (Vaswani et al., 2017), whose complexity grows as the square of the sequence length, SSMs operate linearly, allowing efficient processing of exceptionally long sequences (Gu et al., 2020; 2022b; Smith et al., 2023). Mamba allows

parameters within SSM layers to dynamically vary based on the input sequence (Gu & Dao, 2023), while Mamba-2 further introduces State Space Duality (SSD) to improve computational parallelism and hardware utilization (Dao & Gu, 2024; Waleffe et al., 2024). Recent hybrid architectures combining SSMs with Transformers have further demonstrated significant empirical gains, exploiting the complementary strengths of both architectures (Lieber et al., 2024; Glorioso et al., 2024; Patro & Agneeswaran, 2024; Dong et al., 2024; Waleffe et al., 2024).

**Network Pruning in LLMs.** Network pruning is a widely adopted technique to reduce the computational cost and memory footprint of deep neural networks by eliminating redundant parameters (LeCun et al., 1989; Han et al., 2016). Applying pruning techniques to Large Language Models (LLMs) presents unique challenges compared to smaller models like convolutional networks (Singh & Alistarh, 2020; Benbaki et al., 2023) or even moderate-sized language models like BERT (Devlin et al., 2019). To address these challenges, several efficient pruning methods for LLMs have been proposed. Regarding granularity, these pruning methods for LLMs can be either unstructured, targeting individual weights (Frantar & Alistarh, 2023; Sun et al., 2023; van der Ouderaa et al., 2024), or structured, removing entire units like channels, filters, or attention heads (Ma et al., 2023; Tang et al., 2025). Our work mainly focuses on one-shot unstructured pruning due to its efficiency and potential for high sparsity, while it can also be extended to structured patterns.

**Layer-wise Unstructured Pruning Methods.** To date, layer-wise pruning methods for LLMs are primally based on the optimal brain surgeon (OBS) (Hassibi & Stork, 1992) framework. OBC (Frantar & Alistarh, 2022) proposed the ExactOBS algorithm to reduce computational burden, reformulating layer-wise pruning as a row-wise operation. To address the massive parameters of LLMs, SparseGPT (Frantar & Alistarh, 2023) further tackles the expensive Hessian computation by employing partial weight updates and adaptive mask selection. Other techniques explored more aggressive Hessian estimation (Sun et al., 2023), extension to structured sparsity (Ling et al., 2024; Wei et al., 2024), and methodological improvement for performance (Wu et al., 2024; Yu et al., 2022; Meng et al., 2024).

**Pruning Methods for Mamba.** While pruning algorithms tailored for Transformer-based LLMs have achieved considerable success, pruning Mamba architectures (Gu & Dao, 2023; Dao & Gu, 2024) still encounter substantial challenges. Gwak et al. reveal the redundancy and compressibility of state space models, thereby motivating the application of pruning techniques to SSM architectures (Gwak et al., 2024). Some early studies have focused on structured pruning of Mamba, such as the coarse-grained removal of SSM modules or whole blocks by Mamba-Shedder (Munoz et al., 2025) and on unstructured pruning, evaluating a variety of pruning techniques applied to the Mamba architecture (Ghattas et al., 2025). Nearly, Taghibakhshi et al. propose a group-aware pruning strategy tailored for hybrid attention-SSM models (Taghibakhshi et al., 2025). While Shihab et al. introduce an unstructured pruning framework for Mamba that novelly uses an iterative pruning schedule(Shihab et al., 2025). Compared to earlier strategies, our solution departs in two critical aspects: **(i)** *SparseSSM* extends the classic OBS framework to address pruning in the SSM module, providing rigorous theoretical justification and comprehensive experimental validation. **(ii)** We propose a one-shot, unstructured pruning algorithm for Mamba that requires no fine-tuning.

## 3. Method

In this section, we demonstrate how the OBS framework can be adapted to the Mamba architecture and present our method, *SparseSSM*. To start, we provide a detailed overview of Mamba's forward-propagation pipeline, emphasizing the internal computations within its SSM modules. We then describe our targeted Hessian-matrix calculation technique and derive the resulting importance metrics. Finally, we explore pruning strategies for the feed-forward network (FFN) layers.

### 3.1. Forward Propagation Pipeline of Mamba

We first dive deeper into the forward propagation of a single mamba block. The Mamba architecture decomposes into two complementary components: a feed-forward network (FFN) that performs feature projection and preliminary transformation, and a state space model (SSM) that selectively captures and processes sequential dependencies.

State space models provide a sequence modeling paradigm based on latent state dynamics. In Mamba's SSM layer, the state $\mathbf{h}_t \in \mathbb{R}^{B \times D \times N}$ evolves recurrently with input $\mathbf{x}$ as:

$$\mathbf{h}_t := \widehat{A}\mathbf{h}_{t-1} + \widehat{B}\mathbf{x}_t, \quad \mathbf{y}_t := C^\top \mathbf{h}_t, \qquad (1)$$

where $\widehat{A}$ denotes the state transition matrix , $\widehat{B}$ and $C$ are parameters of SSM, and $\mathbf{y}_t$ is the output. The DSS model (Gu et al., 2022a) first achieves efficient sequence processing by making the pre-discretization SSM input parameters $A, B'$ and $C$ diagonal, then Mamba achieves selectivity by making the diagonal weight matrices time-dependent. Specifically, the original input and output gate matrices $B'$ and $C$ are expanded to shape $\mathbb{R}^{B \times L \times N}$, and the transition parameter $A$ is held via zero-order-hold and discretized into $\Delta A \in \mathbb{R}^{B \times L \times D \times N}$, where $B, L, D, N$ respectively denote the batch size, sequence length, hidden dimension, and latent state space dimension. The discretization and parame-

terization procedures for $A$ are, respectively, as follows:

$$(\Delta A)_{b,\ell,d,n} = \exp\big(\delta_{b,\ell,d}\, A_{d,n}\big),$$
$$(A_{\log})_{d,n} = -\log\big(A_{d,n}\big), \qquad (2)$$

where $\delta_{b,\ell,d}$ denotes element of the stride $\Delta \in \mathbb{R}^{B\times L\times D}$.

This selective, input-dependent design addresses the limitations of earlier linear time-invariant SSMs and enables long-context reasoning by dynamically controlling which state dimensions carry information. However, its recurrent structure and discretization operations render prior methods inapplicable to pruning the parameters of the SSM, specifically the $A$ matrix.

In the Mamba layer, we leverage the selective scan algorithm to traverse the sequence and record the internal state contributions. At each token $t$, we denote by $\mathbf{h}_t \in \mathbb{R}^{B\times D\times N}$ the hidden state tensor at time step $t$, containing the activation values for every batch and channel. This internal signal $\mathbf{h}_t$ reflects how much each state dimension $i$ remembers its past activation at step $t$. By collecting these values across all time steps $t = 1 \dots L$ for a given layer, we obtain token-wise activation statistics for each state dimension. In this way, the selective scan provides a direct window into the layer's memory utilization, which we will exploit to guide pruning.

### 3.2. Hessian Matrix Estimation of SSM Layer

To formally quantify each parameter's importance in SSM layers, we adopt a Hessian-based analysis inspired by optimal brain surgeon (OBS) pruning (Hassibi & Stork, 1992; Frantar & Alistarh, 2023). The goal of pruning is to identify a sparse weight matrix $A_{log}$ that minimizes the reconstruction error between the original and pruned layer outputs. Let **SSM** denote Mamba's SSM layer, then the problem can be formulated as:

$$\arg\min_{A'} \big\|\mathbf{SSM}\big(A_{log},\theta,x\big) - \mathbf{SSM}\big(A_p,\theta,x\big)\big\|_2^2, \quad (3)$$

where $\theta$ denotes the output of the formal linear projection and $A_p$ denotes the pruned weight matrix. Based on OBS framework, pruning of parameter $A_{log}$ requires no compensation because it is essentially a concatenation of multiple diagonal matrices, whose elements' importance can be defined as $\varepsilon_m = w_m^2 H_{mm}$, where $H_{mm}$ denotes the $m$-th diagonal element of the Hessian matrix. It actually measures the curvature of the loss concerning the parameter $m$. However, precisely computing the entries of the Hessian matrix for an SSM module is challenging, because for a given input x, the SSM may be unrolled across time steps

as follows:

$$h_i = \Delta A_i \odot h_{i-1} + \widehat{B}_i \odot x\,, \quad h_{-1} = 0\,,$$
$$y_i = h_i^\top C_i = \sum_{j=0}^{i}\Big[\big(\prod_{k=j+1}^{i}\Delta A_k\big)\odot \widehat{B}_j \odot x_j\Big]^\top C_i\,, \quad (4)$$

where $x$ denotes the input of the SSM module, and $y_i$ denotes the output at each time step $i$. To address this problem shown in Eq. (4), we consider computing the Hessian matrix at each time step and using the hidden state to assess the importance of elements. For the SSM module, we consider the loss $\mathcal{Loss}_t$ incurred when it processes input data of time-step $t$.

$$\mathcal{Loss}_t = \frac{1}{B}\sum_{b=1}^{B}\ell oss(\mathbf{y}_{b,t}) = \frac{1}{B}\sum_{b=1}^{B}\|\mathbf{y}_{b,t} - \hat{\mathbf{y}}_{b,t}\|_2^2, \quad (5)$$

where $y$ denotes the output of one single time step. Based on this hypothesis, we propose Theorem 3.1 to give an estimation of the Hessian matrix, which, to the best of our knowledge, represents a novel theoretical perspective for analyzing and guiding SSM pruning.

**Theorem 3.1** (SSM Hessian Matrix Estimation). *Let $A_{\log} \in \mathbb{R}^{D\times N}$ be the matrix of parameters for SSM diagonals, $\Delta A \in \mathbb{R}^{B\times L\times D\times N}$ the discretized parameter matrix of $A_{\log}$, and $h_{b,i-1,d,n}$ the hidden activations before step $i$. Under the diagonal character of parameter $A_{log}$, the per-parameter importance score of time step $t$ is:*

$$I_{d,n}^t = \sum_b \Delta A_{b,t,d,n}^2\, h_{b,t-1,d,n}^2 \qquad (6)$$

The complete proof of Theorem 3.1 is provided in Appendix A.

### 3.3. Importance Estimation for Integrated Time Steps

Our proposed Theorem 3.1 enables the Hessian matrix of the SSM module to be quickly and accurately estimated. However, pruning $A_{log}$ remains challenging due to its parameter time-sharing property. This implies that while the activation at each time step can produce a pruning mask for $A_{log}$, pruning at one time step affects the selection of the pruning mask at the subsequent time step. Consequently, merging these masks becomes problematic.

To merge these pruning masks produced by each time step, we propose a hierarchical aggregation protocol. To reconcile the uneven importance of different time steps, we aggregate step-wise saliency with an appropriate weighting and selection scheme. Empirically, in Mamba's selective scan, We logged the hidden state at every time step and computed the expectations $\mathbb{E}\|h_t\|_2^2$ across a held-out set of inputs. As illustrated in figure 2, $\mathbb{E}\|h_t\|_2^2$ exhibits a highly consistent pattern across layers. Consequently, most of the state

**Algorithm 1** Time-Selective One-Shot OBS Pruning for the SSM Matrix $A_{\log}$

---

1: **Phase 1: Statistic accumulation**
2: $\mathbf{S} \leftarrow \mathbf{0}_{L \times D \times N}, \; n \leftarrow 0$
3: **for** each sample-batch $N_s = 1, \ldots, n_s$ **do**
4:    Run the forward pass and collect $\mathbf{h}_{1:L}^{(n_s)}$
5:    $n \leftarrow n + 1$
6:    $\mathbf{S} \leftarrow \dfrac{n-1}{n}\mathbf{S} + \dfrac{1}{n}\big(\mathbf{h}_{1:L}^{(n_s)}\big)^2$
7: **end for**
8: **Phase 2: Time-weighted global scoring**
9: $K \leftarrow \lceil pDN \rceil$
10: $w_t \leftarrow (t+1)^{-\mu} \quad$ for $t = 1, \ldots, L$
11: $w_t \leftarrow \dfrac{w_t}{\sum_{u=1}^{L} w_u}$ // *prefix-biased weights*
12: $\mathbf{Q} \leftarrow \sum_{t=1}^{L} w_t\Big(\boldsymbol{\Delta A}_t^2 \odot \mathbf{S}_t\Big)$
13: $\mathcal{I}_\star \leftarrow \mathrm{TopK}_{\mathrm{smallest}}(\mathbf{Q}, K)$
14: $\mathbf{Mask} \leftarrow \mathbf{1}_{D \times N}$
15: $\mathbf{Mask}_{\mathcal{I}_\star} \leftarrow 0$
16: $\widetilde{A}_{\log} \leftarrow A_{\log} \odot \mathbf{Mask}$

---

information is rapidly accumulated in the early timesteps, and **prefix information** deserves higher emphasis. At the same time, very early measurements can be transient and noisy, and the effective influence should decay with time. We therefore adopt a power-law temporal weighting that is heavy-tailed yet monotone,

$$w_t \propto (t+1)^{-\mu}, \qquad \mu > 0, \tag{7}$$

where $\mu$ denotes the power-law hyperparameter controlling the decay rate. It privileges early steps without overly collapsing mass onto the first few tokens. This prefix-biased, power-law aggregation consistently improves pruning decisions for time-shared parameters such as $A_{log}$. Algorithm 1 provides the detailed steps and explanation of our proposed *SparseSSM* method.

**Algorithm Overview.** The Algorithm consists of two stages: statistic accumulation and time-weighted global scoring, which is presented under the single-sample setting (batch size = 1). In Phase 1, we estimate the diagonal Hessian statistics associated with the SSM transition parameter $A_{\log}$. For each calibration mini-batch, we run a forward pass and collect the hidden states $h_{1:L}^{(b)}$ across all timesteps. Following the implementation of SparseGPT (Frantar & Alistarh, 2023), we average the hidden states at each timestep across the sample times $N_s$. In Phase 2, we aggregate timestep saliency into a single global pruning score. We compute the importance score of each element using the previously derived formulation, followed by timestep weighting and normalization. Finally, *SparseSSM* prunes the $K = \lceil sDN \rceil$ entries with the smallest global saliency scores and applies the resulting binary mask to $A_{\log}$.

### 3.4. Sensitivity-Aware Pruning of the FFN Component

Since FFNs account for the majority of parameters, we also prune the standard feed-forward networks (FFNs) in the model to further reduce the parameter count. The forward-propagation blocks in Mamba are composed primarily of linear layers and one-dimensional convolutional layers. Inspired by (Shao et al., 2024), we conducted a module-wise pruning analysis within the feed-forward network (FFN) and found that their pruneability varies substantially (see Appendix B.2.3). In particular, the in_proj and out_proj modules require more careful pruning, as they exhibit higher sensitivity. Moreover, we empirically observe that the reconstruction error of each module grows as its Hessian trace increases, with the rate of this growth varying across modules, as shown in figure 3.

Motivated by these findings, we adopt the sensitivity-aware pruning framework, treating the in_proj and out_proj modules independently. We use the trace of the Hessian matrix of the weights as the sensitivity score, and define the sparsity ratio as:

$$sparsity = 1 - p - \alpha + \frac{2\alpha\, id}{N-1}, \tag{8}$$

where $p$ is the target global sparsity, $\alpha$ defines the allowable deviation interval $[1 - p - \alpha, \; 1 - p + \alpha]$, $N$ is the total number of weights, and $id$ is the sensitivity-rank index of the given weight after sorting by Hessian-trace importance. Here, we adopt the sparsity law computation from (Shao et al., 2024). However, the redundancy analysis for the Mamba architecture is more important and constitutes the main novelty of this part. This formulation ensures that higher-sensitivity modules (larger $id$) are assigned lower sparsity levels, while exactly satisfying the overall sparsity budget $p$.

## 4. Experiments

In this section, we benchmark *SparseSSM* against leading pruning algorithms for Mamba-based LLMs. The complete experimental protocol and reproducibility details appear in Appendix B.1.

**Models and Datasets.** We evaluate *SparseSSM* on the public Mamba-1 checkpoints ranging from 130 million to 2.8 billion parameters (Gu & Dao, 2023), and Mamba-2 checkpoints ranging from 370 million to 2.7 billion parameters. Additionally, we include the 8B-parameter Mamba-2 model trained with the Megatron framework and released as part of the empirical study of Mamba-based LLMs, in order to stress-test the robustness and scalability of our method on a large production-scale model. For all models, we follow the standard calibration protocol on WikiText-2: we randomly sample 128 contiguous segments of 2048 tokens each from the first data shard, as in (Frantar & Alistarh,

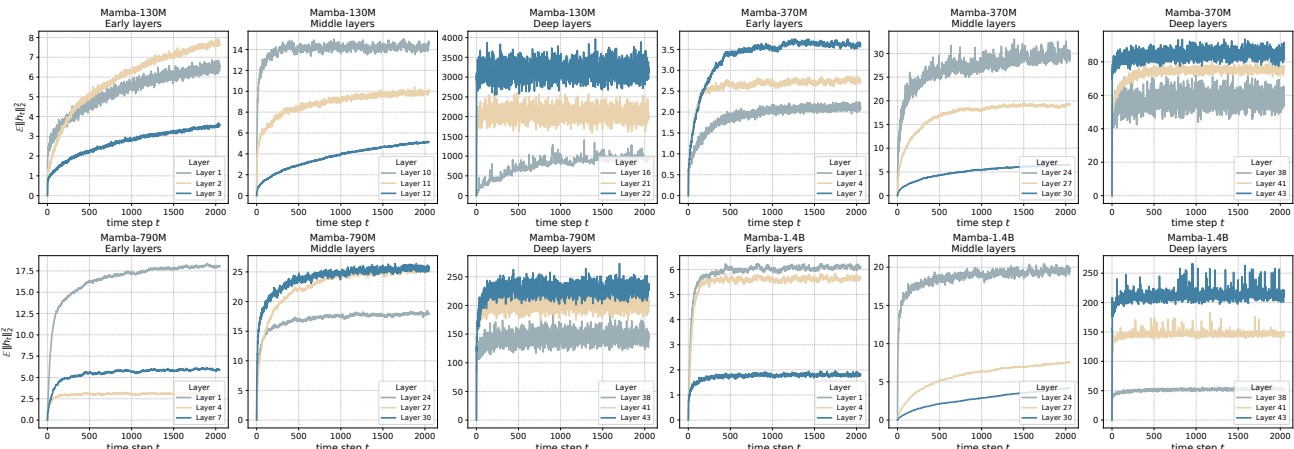

Figure 2. The evolution of $H_t$ over timesteps across early, middle, and deep layers in different Mamba model variants.

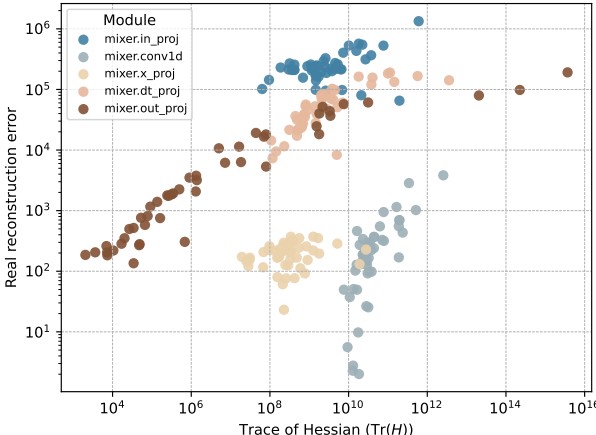

Figure 3. The trace of the Hessian matrices and the real reconstruction errors for each Mamba-370M FFN module across different layers at a sparsity level of 50%.

2023). Perplexity is computed as the exponential of the negative log-likelihood per token, consistent with (Hugging Face). Downstream evaluation uses the raw WikiText-2 validation set (Merity et al., 2017), the Penn Treebank corpus (Marcus et al., 1994), and a 10000-document slice of the C4 validation split (Raffel et al., 2020). In addition, to probe behavior on a modern, information-dense web corpus, we report supplementary results on a validation subset of the FineWeb dataset (Penedo et al., 2024) in Appendix B.1. Zero-shot generalization is measured on PIQA (Bisk et al., 2020), OpenBookQA (Mihaylov et al., 2018), Winogrande (Sakaguchi et al., 2021), ARC-Easy, and ARC-Challenge (Clark et al., 2018) without any fine-tuning. This suite covers both language modeling and reasoning benchmarks, allowing a comprehensive assessment of model performance and generalization. During implementation, we also referred to mamba-minimal (Ma) for guidance.

**Baselines.** We compare *SparseSSM* against three repre-

sentative pruning methods under identical calibration and sparsity budgets. First, global magnitude pruning follows the classical heuristic of removing the smallest-magnitude weights (Han et al., 2015). Second, SparseGPT applies a Hessian-aware one-shot pruning strategy (Frantar & Alistarh, 2023). However, SparseGPT is not inherently suited to the structural characteristics of SSM modules. Here, we present the results of its naive application. Third, Mamba-Shedder is a recent selective state space variant tailored for Mamba architectures (Munoz et al., 2025). Additionally, we also include Wanda (Sun et al., 2023) as an extra baseline, which is a recent pruning method developed under the OBS framework. *All baselines and our method share the same configuration to ensure fairness.*

### 4.1. Results of Pruning SSM Modules

We first isolate the SSM blocks and prune only the learnable diagonal $A_{log}$ matrices. Within the state-space module (SSM), Mamba reparameterizes $A$ via its negative logarithm to enforce $A < 0$, thus preserving the module's robustness. Indeed, the parameter $A_{log}$ plays a role analogous to the forget gate in LSTM (Hochreiter & Schmidhuber, 1997) networks and has a profound impact on the predictive capacity of the language model.

Table 1 reports detailed token-level perplexity and zero-shot accuracies at 50% sparsity. As demonstrated, our pruning strategy shows excellent efficacy in maintaining SSM stability. Across all model scales, *SparseSSM* consistently outperforms MP, Mamba-Shedder, and SparseGPT at the same sparsity, matching or exceeding the dense baseline on most zero-shot tasks while retaining competitive perplexities. For example, on the Mamba-370M model, our method attains no degradation on the majority of zero-shot evaluations and yields a ~5.4% average accuracy gain over alternative pruners, without any fine-tuning. The gains stem from our second-order importance metric combined with a

time-weighted mask, as detailed in Section 3.

Additionally, to assess pruning effectiveness at higher sparsity, we evaluate models of different sizes across a range of sparsity levels and examine their performance, see Appendix B.2.9. We provide additional results on more baselines, datasets, and model scales in the Appendix B.2.6.

### 4.2. Results of Pruning the Whole Mamba Architecture

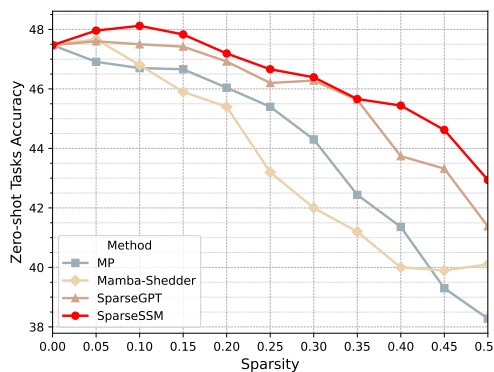

*Figure 4.* Performance of the full Mamba architecture at multiple sparsity levels by measuring zero-shot task accuracy

We then apply one-shot unstructured pruning across all trainable weights except the input embedding and output head. In this setting, each model typically incorporates an `nn.Conv1d` layer for feature preprocessing, `in_proj` and `out_proj` linear layers for dimensionality transformation, and immediately before the selective scan operation, a learnable `x_proj` mapping that produces the parameters $\Delta, B, C$, concurrently, the temporal stride parameter $\Delta$ is reparameterized via `dt_proj`. Empirical analysis reveals that these modules exhibit markedly heterogeneous pruning tolerances: pruning of the `in_proj` and `out_proj` layers induces substantially larger degradations than other linear modules, detailed comparison results are in Appendix B.2.3.

However, when we jointly prune both the SSM modules and the FFN branches, our proposed method *SparseSSM* outperforms all baselines, achieving lower perplexity and higher zero-shot accuracy across every model scale, as shown in Table 5 in Appendix B.2.1. Figure 4 further illustrates our method's performance across multiple sparsity levels. As demonstrated, for each downstream task, the pruned models exhibit consistent improvements, with gains becoming especially pronounced under higher sparsity regimes.

### 4.3. Results of Semi-Structure, Structure Sparsity Extension and Speedup

Our approach admits a straightforward extension to $N : M$ and fully structured pruning. In fact, during unstructured pruning experiments, we observed that the pruned entries in

the parameter $A_{log}$ overwhelmingly cluster within particular columns. Certain hidden state channels in the state space model exhibit markedly higher redundancy. This empirical finding underpins the strong performance of our structured pruning scheme.

Table 3 reports results on the Mamba-370M model under 2:4 and 4:8 sparsity patterns. At the same overall sparsity, our method delivers smaller performance degradation in $N : M$ pruning.

Without specialized acceleration kernels, unstructured pruning does not directly translate into wall-clock speedup. Therefore, we extend our method to the structured pruning setting, where the reduced model dimensions enable practical acceleration to be directly observed. To implement structured pruning, we target the second axis of $A_{log}$. We employ the weight itself to guide the pruning and aggregate the importance of each column by computing its $L_1$ norm. Simultaneously, we resize the output dimension of the linear `x_proj` layer. As shown in Table 4, this structured pruning on Mamba-370M induces only negligible accuracy loss at 50% sparsity without any fine-tuning, while accelerating the SSM module by a factor of $1.72\times$. Although the number of parameters in $A_{log}$ is relatively small, its contribution to acceleration is substantial because it is repeatedly utilized.

We perform one warm-up forward pass to amortize one-time compilation overheads, and then a second forward pass using CUDA events to test the speedups in recurrent mode. As reported in Table 2, our pruning method yields consistent runtime acceleration across different Mamba model variants. Therefore, pruning directly translates into significant end-to-end runtime gains in practical inference scenarios. Detailed configurations are provided in the Appendix B.2.4. Meanwhile, we also measure the acceleration after structured pruning on a decoding task, as well as the speedup of the selective scan operator. The results are reported in the Appendix B.2.4. In addition, our method also leads to substantial memory savings, see Appendix B.2.5 for memory results.

### 4.4. Generalize to other SSM architecture

We conduct experiments on Mamba-2 models of various scales, including Mamba-2-8B (Waleffe et al., 2024), with the results reported in the Appendix B.2.6.

Beyond the vanilla Mamba language models considered in the above experiments, recent work has extended SSMs to diverse domains, including vision backbones and large-scale hybrid Transformer–Mamba language models.

MambaVision (Hatamizadeh & Kautz, 2025) is a recent vision backbone that redesigns the Mamba formulation for 2D feature maps and interleaves SSM blocks with ViT-style attention. We adopt the official pretrained checkpoints re-

*Table 1.* Performance analysis for one-shot unstructured pruning of $A_{log}$ matrix of SSM modules in Mamba models at $50\%$ sparsity. Here, ↓ lower metrics reflect better outcomes, and ↑ denotes higher metrics reflect better outcomes.

| Model | Method | Wiki. ↓ | PTB ↓ | C4 ↓ | OBQA ↑ | PIQA ↑ | ARC-e ↑ | ARC-c ↑ | WinoG ↑ | Avg. ↑ |
|---|---|---|---|---|---|---|---|---|---|---|
| Mamba-130M | Dense | 20.60 | 32.75 | 25.66 | 28.60 | 63.28 | 48.02 | 24.40 | 52.5 | 43.36 |
| | MP (Han et al., 2015) | 740.3 | 1109 | 273.0 | 26.80 | 58.05 | 39.69 | 22.35 | **52.33** | 39.84 |
| | Wanda (Sun et al., 2023) | 77.84 | 111.7 | 52.32 | 28.60 | 60.88 | 39.69 | 23.21 | 51.78 | 40.83 |
| | Mamba-Shedder (Munoz et al., 2025) | 698.7 | 1544 | 532.6 | 28.00 | 54.73 | 30.00 | 23.72 | 49.88 | 37.27 |
| | SparseGPT (Frantar & Alistarh, 2023) | 2.4e7 | 6.1e6 | 3.9e5 | 27.60 | 55.28 | 30.64 | 23.98 | 49.25 | 37.35 |
| | *SparseSSM* | **21.26** | **33.82** | **26.31** | **30.60** | **63.33** | 45.62 | **24.66** | 50.75 | **42.99** |
| Mamba-370M | Dense | 14.32 | 23.46 | 19.37 | 31.00 | 68.34 | 54.97 | 27.90 | 55.25 | 47.49 |
| | MP (Han et al., 2015) | 291.2 | 535.9 | 105.0 | **30.40** | 61.70 | 44.23 | 22.61 | 51.38 | 42.06 |
| | Wanda (Sun et al., 2023) | 40.91 | 79.89 | 37.49 | 27.60 | 64.20 | 37.71 | 25.85 | 53.35 | 41.74 |
| | Mamba-Shedder (Munoz et al., 2025) | 334.5 | 446.6 | 221.81 | 23.80 | 54.19 | 29.42 | 23.12 | 52.25 | 36.56 |
| | SparseGPT (Frantar & Alistarh, 2023) | 2696 | 7570 | 613.2 | **30.40** | 65.23 | 49.16 | 25.60 | 52.41 | 44.56 |
| | *SparseSSM* | **15.04** | **24.63** | **20.13** | 30.20 | 68.34 | 55.26 | 27.90 | 55.64 | 47.47 |
| Mamba-790M | Dense | 11.96 | 18.45 | 16.62 | 33.80 | 72.63 | 61.07 | 29.44 | 56.27 | 50.64 |
| | MP (Han et al., 2015) | 179.0 | 377.0 | 79.43 | 30.20 | 64.74 | 47.81 | 25.09 | 54.14 | 44.40 |
| | Wanda (Sun et al., 2023) | 27.63 | 50.45 | 27.83 | 31.40 | 66.00 | 52.02 | 26.19 | 53.04 | 45.73 |
| | Mamba-Shedder (Munoz et al., 2025) | 225.48 | 256.32 | 195.47 | 28.20 | 56.47 | 33.29 | 21.50 | 51.07 | 38.11 |
| | SparseGPT (Frantar & Alistarh, 2023) | 110.5 | 242.19 | 81.87 | **32.80** | 68.34 | 54.42 | 27.47 | 54.93 | 47.59 |
| | *SparseSSM* | **12.47** | **19.43** | **17.19** | 32.60 | 70.40 | 57.11 | 30.46 | 56.27 | 49.37 |
| Mamba-1.4B | Dense | 10.75 | 17.05 | 15.17 | 36.40 | 73.88 | 65.57 | 32.85 | 61.17 | 53.98 |
| | MP (Han et al., 2015) | 100.7 | 190.8 | 54.49 | 30.60 | 67.95 | 53.28 | 24.06 | 52.49 | 45.68 |
| | Wanda (Sun et al., 2023) | 21.17 | 38.19 | 21.86 | 32.20 | 68.55 | 56.69 | 27.73 | 51.62 | 47.36 |
| | Mamba-Shedder (Munoz et al., 2025) | 223.1 | 293.7 | 190.5 | 27.20 | 56.86 | 34.09 | 23.04 | 51.46 | 38.53 |
| | SparseGPT (Frantar & Alistarh, 2023) | 49.77 | 88.20 | 40.74 | 34.40 | 71.38 | 60.10 | 30.03 | 54.78 | 50.14 |
| | *SparseSSM* | **11.89** | **18.93** | **16.49** | 35.00 | 72.42 | 61.99 | 31.23 | 56.83 | 51.49 |
| Mamba-2.8B | Dense | 9.45 | 14.79 | 13.61 | 39.60 | 75.84 | 69.70 | 36.35 | 63.06 | 56.91 |
| | MP (Han et al., 2015) | 61.08 | 70.75 | 39.13 | **35.40** | 70.57 | 59.13 | 29.35 | 60.30 | 50.95 |
| | Wanda (Sun et al., 2023) | 15.31 | 21.10 | 17.53 | 34.40 | 72.36 | 63.05 | 30.12 | 57.93 | 51.57 |
| | Mamba-Shedder (Munoz et al., 2025) | 104.7 | 126.6 | 112.1 | 25.80 | 59.00 | 40.20 | 23.50 | 51.20 | 39.95 |
| | SparseGPT (Frantar & Alistarh, 2023) | 56.83 | 92.31 | 44.43 | 34.80 | **73.94** | 65.74 | **34.04** | 58.56 | 53.42 |
| | *SparseSSM* | **10.63** | **16.98** | **15.14** | **35.40** | 73.45 | 65.07 | 33.45 | 60.54 | **53.58** |

*Table 2.* End-to-end measured speedups on a single GPU before and after pruning 50% parameters of the state space module across different model variants.

| Model | Dense (ms) | Pruned (ms) | Speedup |
|---|---|---|---|
| Mamba-130m | 251.4 (± 0.20) | 232.5 (± 0.18) | 1.08 (± 0.00)× |
| Mamba-370m | 546.7 (± 0.35) | 329.4 (± 0.27) | 1.66 (± 0.00)× |
| Mamba-790m | 661.9 (± 0.64) | 388.5 (± 0.51) | 1.70 (± 0.00)× |
| Mamba-1.4b | 1090 (± 0.02) | 656.1 (± 1.39) | 1.66 (± 0.00)× |
| Mamba-2.8b | 1648 (± 0.04) | 737.6 (± 2.31) | 2.23 (± 0.01)× |

*Table 3.* Performance analysis for one-shot pruning of the SSM module in Mamba-370M at 2 : 4 and 4 : 8 sparsity patterns. Here, Avg. denotes the average accuracy over five selected zero-shot tasks.

| Sparsity | Method | Wiki. ↓ | PTB ↓ | C4 ↓ | Avg. ↑ |
|---|---|---|---|---|---|
| 2 : 4 | MP | 77.20 | 135.9 | 59.74 | 38.00 |
| | *SparseSSM* | **16.90** | **26.91** | **21.61** | **46.50** |
| 4 : 8 | MP | 81.56 | 148.25 | 63.76 | 38.19 |
| | *SparseSSM* | **16.56** | **26.44** | **21.40** | **46.83** |

*Table 4.* Performance analysis for one-shot structured pruning of the SSM module in Mamba-370M. Here, Avg. denotes the average accuracy over five selected zero-shot tasks.

| Sparsity | Method | Wiki. ↓ | PTB ↓ | C4 ↓ | Avg. ↑ |
|---|---|---|---|---|---|
| 25% | MP | 35.27 | 71.12 | 33.85 | 41.49 |
| | *SparseSSM* | **15.22** | **24.80** | **20.38** | **46.85** |
| 50% | MP | 117.0 | 162.7 | 66.74 | 37.98 |
| | *SparseSSM* | **18.13** | **28.82** | **22.65** | **46.31** |

(Lieber et al., 2024) interleaves Transformer attention layers with Mamba SSM layers and incorporates sparse Mixture-of-Experts blocks. The pruned Jamba models retain most of the baseline performance. The detailed experiment setting and results are in the Appendix B.2.7.

### 4.5. Ablation Study

We conduct a systematic ablation study on the key components of *SparseSSM* to isolate their contributions to pruning efficacy. In particular, we find that accurate Hessian matrix estimation is instrumental to our method's superior performance, while incorporating a weighted strategy yields additional gains. As shown in Figure 5, our full strategy

leased by NVLabs and directly apply our pruning algorithm to the SSM modules. Empirically, the pruned MambaVision models largely preserve the original accuracy. Jamba

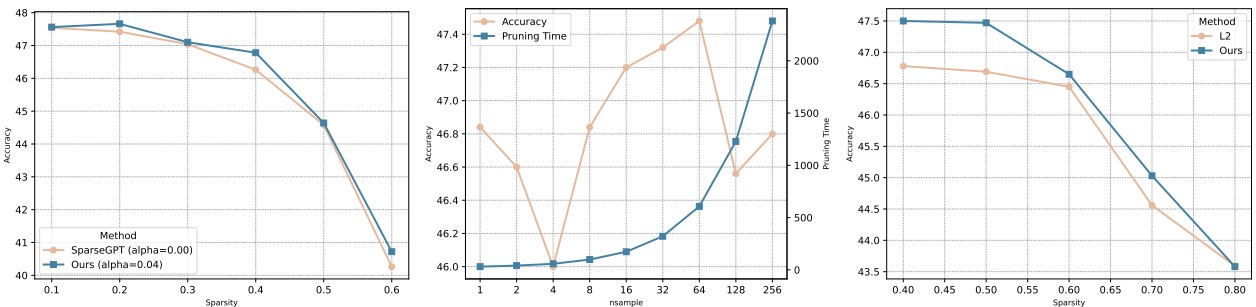

*Figure 5.* Effects of calibration sample size, sparsity interval, and aggregation method. **(Left)** shows a performance analysis of pruning the FFN components of Mamba-370M under varying sparsity settings, **(middle)** shows a performance analysis and pruning efficiency analysis of pruning the SSM modules as the calibration sample size is varied, while **(right)** shows a performance analysis of different aggregation methods.

significantly outperforms a simpler baseline that applies our Hessian estimate via a simple $L_2$ norm aggregation over time steps.

We also assess the effects of sensitivity pruning width and calibration data volume on final results. We change the super parameter $\alpha$ that controls pruning width and $N_{sample}$ that controls calibration data volume. Our experiments reveal that selecting an appropriately sized width parameter substantially improves pruning outcomes in the FFN components, surpassing the performance of SparseGPT. As for calibration data, we observe that fewer than 16 samples degrade the performance of the pruned model. However, a sampling count of 64 strikes the best trade-off between pruning quality and computational efficiency. More ablation studies on pruning the SSM and FFN separately, the calibration datasets used for pruning, the power-law hyperparameter $\mu$, and different weighting strategies are provided in the Appendix B.2.8.

## 5. Conclusion

In this work, we introduce *SparseSSM*, a one-shot, training-free unstructured pruning framework that extends the classic OBS paradigm to selective state-space modules in Mamba-based LLMs. By incorporating time-sharing parameter saliency and explicitly accounting for the discretization of the state-transition matrix, our layer-wise algorithm computes local second-order importance scores and reconstructs remaining weights to minimize output error. Furthermore, our module sensitivity analysis reveals distinct pruning tolerances between input and output projections, offering new insights into redundancy within state-space architectures. Our results establish that state-space LLMs like Mamba can be compressed as effectively as their Transformer counterparts via principled, OBS-guided pruning, paving the way for more efficient deployment within resource-restricted contexts. In future work, we plan to further extend *SparseSSM*

to structured pruning of the entire Mamba architecture. We also aim to generalize our time weighted aggregation and insights on time-shared parameters to other time-varying architectures and investigate hardware-aware optimizations that further accelerate sparse state-space inference.

## Impact Statement

Our proposed method effectively reduces parameter redundancy in Mamba-based LLMs, yielding a leaner network representation that requires fewer floating-point operations during inference. As a result, these pruned models can be deployed with lower computational cost, both in terms of GPU hours and energy consumption, thereby democratizing access to state-of-the-art LLM capabilities for academic, industrial, and edge computing environments. Moreover, by curtailing the extensive resource demands traditionally associated with LLM inference, our approach contributes to a reduction in the cumulative electricity usage and associated carbon emissions of LLM workloads. In doing so, it supports the broader agenda of sustainable AI by mitigating the environmental and climate impacts of deploying LLMs at scale.

## Acknowledgement

This paper is supported by Young Scientists Fund of the National Natural Science Foundation of China (NSFC) (No. 62506305), Zhejiang Leading Innovative and Entrepreneur Team Introduction Program (No. 2024R01007), Key Research and Development Program of Zhejiang Province (No. 2025C01026), Scientific Research Project of Westlake University (No. WU2025WF003), Chinese Association for Artificial Intelligence (CAAI) & Ant Group Research Fund - AGI Track (No. 2025CAAI-ANT-13). It is also supported by the research funds of the National Talent Program and Hangzhou Municipal Talent Program.

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

# A. Proofs of Theorem 1

*Proof.* We begin with a trained network's parameters, where $A_{\log}$ is near a local minimum of the loss $\mathcal{L}oss$. In this setting, small perturbations of the parameters cause a loss increase dominated by the quadratic term of the second-order Taylor expansion.

**Lemma A.1** (OBS Parameter Importance). *Under the second-order OBS pruning framework, let $\mathbf{H} = \nabla^2 \mathbf{L}(\theta)$ denote the Hessian of the loss $L$ with respect to the full parameter vector $\theta$. Then the saliency of the individual parameter $\mathbf{\Delta A_{t,d,n}}$ is given by*

$$I_{d,n}^t \;=\; \frac{\left(\Delta A_{t,d,n}\right)^2}{2\left[H^{-1}\right]_{(d,n),(d,n)}} \;=\; \frac{1}{2}\,H_{(d,n),(d,n)}\left(\Delta A_{t,d,n}\right)^2. \tag{9}$$

**Returning to the proof of the main theorem.** We analyze the single scalar coordinate when the batch size is 1

$$w \;\equiv\; \Delta A_{t,d,n}, \tag{10}$$

which multiplies its corresponding hidden coordinate at step $t$:

$$h_{t,d,n} \;=\; w\,h_{t-1,d,n} \;+\; (\Delta B_t u_t)_{d,n}. \tag{11}$$

Consider a per-step mean–squared error (MSE) objective at time $t$:

$$\mathcal{L}oss \;=\; \left\| F_t(w) - T_t \right\|_2^2, \tag{12}$$

where $F_t(w)$ is the network output at step $t$ and $T_t$ is the target. Its gradient with respect to a scalar parameter $w$ is

$$\frac{\partial \mathcal{L}oss}{\partial w} \;=\; \left(\frac{\partial F_t(w)}{\partial w}\right)^{\top}\left(F_t(w) - T_t\right). \tag{13}$$

The exact Hessian is

$$\frac{\partial^2 \mathcal{L}oss}{\partial w^2} \;=\; \left(\frac{\partial F_t(w)}{\partial w}\right)^{\top}\frac{\partial F_t(w)}{\partial w} \;+\; \left(F_t(w) - T_t\right)^{\top}\frac{\partial^2 F_t(w)}{\partial w^2}. \tag{14}$$

Near a trained solution, the residuals $F_t(w) - T_t$ are small. Thus, the diagonal-OBS approximation gives

$$H_{mm} \;\approx\; \left\| \frac{\partial F_t(w)}{\partial w} \right\|_2^2, \tag{15}$$

The Hessian is approximately the sample covariance of the output Jacobians with respect to $w$. At the fixed time step $t$, the coordinate $h_{t,d,n}$ satisfies

$$\Delta A \in \mathbb{R}^{L \times D \times N}, \; h_{t,d,n} \in \mathbb{R}^{L \times D \times N} \tag{16}$$

Since the computation takes the form

$$h_{t,d,n} = w \odot h_{t-1,d,n} + (\Delta B_t u_t)_{d,n}. \tag{17}$$

Here, $u_t$ denotes the input at step $t$, which is fixed with respect to $w$. $\Delta B_t$ depends on the input, but does not depend on the scalar variable $w$. By direct differentiation of the local update,

$$\frac{\partial h_{t,d,n}}{\partial w} \;=\; h_{t-1,d,n}. \tag{18}$$

Therefore,

$$H_{mm} \;\approx\; h_{t-1,d,n}^2 \;\propto\; h_{t-1,d,n}^2, \tag{19}$$

where the proportionality absorbs positive, step-dependent factors independent of $w$.

According to lemma A.1, the OBS saliency for a scalar parameter $w$ is,

$$S(w) \;=\; \frac{w^2}{2\left[H^{-1}\right]_{mm}} \;=\; \frac{1}{2}\,H_{mm}\,w^2, \tag{20}$$

and using the estimate of $H_{mm}$ above with $w = \Delta A_{t,d,n}$, the per-sample contribution satisfies

$$S\big(\Delta A_{t,d,n}\big) \ \propto \ \big(\Delta A_{t,d,n}\big)^2 \, h_{t-1,d,n}^2. \tag{21}$$

Summing over the batch at fixed $(t, d, n)$ yields the per-step importance

$$I_{d,n}^t \ = \ \sum_b \Delta A_{b,t,d,n}^2 \, h_{b,t-1,d,n}^2 \tag{22}$$

which completes the proof. $\qquad\square$

### A.1. Proof of Lemma A.1

Let $\theta$ denote the vector of all parameters and $H = \nabla^2 L(\theta)$ the Hessian at the optimum. For a perturbation $\Delta\theta$, the Taylor expansion gives:

$$\Delta L \ \approx \ \frac{1}{2}\,\Delta\theta^T H \,\Delta\theta\,. \tag{23}$$

In the SSM module, Over a small time increment $\delta_{t,d}$ at step $t$, the state update is:

$$h_{t,d,n} \ = \ \Delta A_{t,d,n} \, h_{t-1,d,n} \ + \ \Delta\big(B_u\big)_t\,, \tag{24}$$

where $\Delta\big(B_u\big)_i$ is independent with parameter $A$. The only way $\Delta A_{t,d,n}$ affects the network's forward pass is through this scalar multiplier $e^{A_{d,n},\delta_{b,i,d}}$ at each time step. Crucially, because $A_{log}$ is diagonal, each parameter $\Delta A_{t,d,n}$ influences only its corresponding state dimension $d$ in SSM $n$, independently of other dimensions, which implies

$$\frac{\partial^2 L}{\partial\Delta A_{t,d,n}\partial\Delta A_{t,d',n'}} \ = \ 0\,, \ \big(d',n'\big) \neq \big(d,n\big). \tag{25}$$

Therefore, the Hessian matrix $H$ has the characteristic

$$\big[H^{-1}\big]_{(d,n),(d,n)} = \frac{1}{H_{(d,n),(d,n)}}. \tag{26}$$

where $H_{(d,n),(d,n)} = \partial^2 L \,/\, \partial\Delta A_{t,d,n}^2$ is the Hessian's diagonal entry for that parameter. Combining with the classic OBS saliency definition $\varepsilon_m = w_m^2 \,/\, [\mathbf{H}^{-1}]_{mm}$ (Hassibi & Stork, 1992), then we define the OBS saliency of parameter $A_{log,d,n}$ as

$$I_{d,n}^t \ = \ \frac{\big(\Delta A_{t,d,n}\big)^2}{2\big[H^{-1}\big]_{(d,n),(d,n)}} \ = \ \frac{1}{2}\,H_{(d,n),(d,n)}\big(\Delta A_{t,d,n}\big)^2. \tag{27}$$

$\qquad\square$

## B. Experiments Details

### B.1. Experiments Setup

We performed all experiments on a dedicated server using dual Intel Xeon Platinum 8457C processors (48 cores / 96 threads each), 512 GB of DDR5 memory, and eight NVIDIA GeForce RTX 4090 GPUs (24 GB each). We used the PyTorch library to implement the Mamba model and pruning methods for our experiments.

We based our implementation on the SparseGPT code framework (Frantar & Alistarh, 2023), performing pruning on a per-module basis by registering forward hooks to capture each module's inputs during the forward pass. After pruning a given layer, we update its inputs to maintain correct activation propagation. For each pruned module, we remove the designated parameters to realize the prescribed sparsity.

In our Mamba implementation, we adopted the mamba-minimal (Ma) code framework and loaded the official Mamba checkpoint (Gu & Dao, 2023) for pretrained weights.

**Hyperparameters.** For SSM-module pruning, we set $N_{sample} = 64$, which we found yields the best trade-off between pruning quality and computational cost, and set $p = 1.0$ for timestep weights. In the FFN pruning stage, we chose $\alpha = 0.04$, implying that each FFN submodule is assigned a sparsity rate of

$$S_{\text{FFN},i} = \begin{cases} 0.96 - p + \dfrac{0.08\,id}{N-1}, & \text{if } i \in \{\texttt{in\_proj}, \texttt{out\_proj}\}, \\ S_{\text{global}}, & \text{otherwise,} \end{cases} \tag{28}$$

where $N$ is the total number of weights, and $id$ is the sensitivity-rank index of the given weight after sorting by Hessian-trace importance. It means that for the modules `in_proj` and `out_proj`, the allowable deviation interval $\left[0.96 - S_{global},\ 1.04 - S_{global}\right]$. For the hierarchical aggregation protocol, we set $p = 1$, which we empirically found to achieve the best performance. The remaining hyperparameters governed the logging and pruning module configuration.

**Implementation Details.** Below, we summarize the precise configurations used for each selected baseline:

- MP (Han et al., 2015): The weight matrix of each module is sorted by absolute value, retaining the $top - k$ entries and zeroing out all others. For SSM modules, the same procedure is applied to the state-transition matrix $A$.
- Mamba-Shedder (Munoz et al., 2025): We employed the authors' published implementation and default settings, without fine-tuning. Since the authors built upon the official Mamba model implementation and introduced their own modifications, we reproduced this baseline by employing the Mamba model definition as provided by the authors.
- SparseGPT (Frantar & Alistarh, 2023): We extended the original `SparseGPT` framework to support Mamba pruning via two adaptations: (1) when pruning `nn.Conv1d` modules, we applied the SparseGPT processing pipeline for `transformer.Conv1d` directly to the `nn.Conv1d` modules; and (2) when pruning the SSM parameter matrix $A$, we enable direct matrix-level pruning and use the hidden state $h$ as calibration data.

## B.2. Additional Experiments Results

### B.2.1. PRUNING THE WHOLE MAMBA ARCHITECTURE

At 30% unstructured sparsity over all trainable weights, *SparseSSM* consistently surpasses MP, Mamba-Shedder, and SparseGPT, yielding lower language–modeling perplexities on WikiText-2, PTB, and C4, and higher zero-shot accuracies on OBQA, PIQA, ARC-E, ARC-C, and Winogrande, see table 5. These gains hold without any fine-tuning and narrow the pruned–vs–dense gap, indicating that our joint treatment of SSM and FFN branches produces robust, scale-transferable improvements.

*Table 5.* Performance analysis for one-shot unstructured pruning of the whole Mamba models (130M ∼ 1.4B) at 30% sparsity. Here, ↓ lower metrics reflect better outcomes, and ↑ denotes higher metrics reflect better outcomes.

| Model | Method | Wiki. ↓ | PTB ↓ | C4 ↓ | OBQA ↑ | PIQA ↑ | ARC-e ↑ | ARC-c ↑ | WinoG ↑ | Avg. ↑ |
|---|---|---|---|---|---|---|---|---|---|---|
| | Dense | 20.60 | 32.75 | 25.66 | 28.60 | 63.28 | 48.02 | 24.40 | 52.5 | 43.36 |
| | MP (Han et al., 2015) | 9.5e5 | 7.0e5 | 3.4e5 | 25.60 | 53.97 | 28.03 | **26.62** | 52.33 | 37.31 |
| Mamba-130M | Mamba-Shedder (Munoz et al., 2025) | 54.59 | 83.88 | 56.58 | 26.80 | 59.74 | 43.60 | 23.04 | 52.41 | 41.12 |
| | SparseGPT (Frantar & Alistarh, 2023) | 646.9 | 1638 | 165.0 | **29.60** | 55.60 | 33.50 | 25.43 | 50.67 | 38.96 |
| | *SparseSSM* | **26.49** | **40.64** | **32.06** | 28.40 | **61.21** | 43.77 | 22.78 | **52.64** | **41.76** |
| | Dense | 14.32 | 23.46 | 19.37 | 31.00 | 68.34 | 54.97 | 27.90 | 55.25 | 47.49 |
| | MP (Han et al., 2015) | 582.4 | 983.3 | 454.2 | 31.20 | 63.60 | 46.10 | 27.00 | 53.60 | 44.30 |
| Mamba-370M | Mamba-Shedder (Munoz et al., 2025) | 40.76 | 61.23 | 41.37 | 29.00 | 62.19 | 46.04 | 22.61 | 49.88 | 41.94 |
| | SparseGPT (Frantar & Alistarh, 2023) | 57.90 | 157.5 | 48.86 | **31.60** | **67.25** | **52.23** | 26.96 | 53.35 | 46.28 |
| | *SparseSSM* | **16.37** | **26.56** | **21.90** | 30.00 | 67.14 | 52.23 | 27.82 | **54.78** | **46.39** |
| | Dense | 11.96 | 18.45 | 16.62 | 33.80 | 72.63 | 61.07 | 29.44 | 56.27 | 50.64 |
| | MP (Han et al., 2015) | 1.9e47 | 5.1e39 | 1.5e43 | 27.00 | 51.74 | 26.18 | 26.28 | 48.86 | 36.01 |
| Mamba-790M | Mamba-Shedder (Munoz et al., 2025) | 29.28 | 44.71 | 33.73 | 29.00 | 65.78 | 50.76 | 25.94 | 53.04 | 44.90 |
| | SparseGPT (Frantar & Alistarh, 2023) | 24.40 | 41.92 | 27.22 | 28.20 | 59.85 | 37.29 | 26.28 | 52.49 | 40.82 |
| | *SparseSSM* | **13.92** | **21.42** | **19.21** | **32.40** | **70.08** | **56.61** | 27.73 | **57.54** | **48.87** |
| | Dense | 10.75 | 17.05 | 15.17 | 36.40 | 73.88 | 65.57 | 32.85 | 61.17 | 53.98 |
| | MP (Han et al., 2015) | 39.61 | 54.20 | 30.87 | 30.80 | 65.51 | 45.71 | 26.71 | 52.17 | 44.18 |
| Mamba-1.4B | Mamba-Shedder (Munoz et al., 2025) | 24.32 | 39.08 | 31.73 | 27.80 | 65.61 | 52.86 | 26.88 | 52.49 | 45.13 |
| | SparseGPT (Frantar & Alistarh, 2023) | 22.72 | 31.18 | 22.75 | 32.20 | 67.95 | 52.69 | 28.50 | 55.49 | 47.37 |
| | *SparseSSM* | **11.61** | **18.52** | **16.25** | **32.60** | **72.42** | **62.46** | **31.40** | **57.93** | **51.36** |

*Table 6.* Performance analysis of pruning time overhead. Specifically, we conduct experiments on multiple model variants and across different calibration-data sample sizes.

| Model | Layers | Hidden size | Nsample | Pruning time |
|---|---|---|---|---|
| Mamba-130M | 24 | 768 | 32 | 164.4378 s |
| | | | 64 | 311.3634 s |
| | | | 128 | 624.6192 s |
| Mamba-370M | 48 | 1024 | 32 | 319.0448 s |
| | | | 64 | 602.5500 s |
| | | | 128 | 1203.028 s |
| Mamba-790M | 48 | 1536 | 32 | 326.1090 s |
| | | | 64 | 630.0898 s |
| | | | 128 | 1239.914 s |
| Mamba-1.4B | 48 | 2048 | 32 | 348.4770 s |
| | | | 64 | 662.2011 s |
| | | | 128 | 1272.2396 s |

### B.2.2. PRUNING EFFICIENCY ANALYSIS

Our proposed method can prune Mamba-based large language models in an extremely short time, as table 6 shows. Specifically, thanks to our efficient Hessian matrix estimation method and fully parallelized implementation, the time required to compute pruning scores is virtually negligible; the primary time overhead instead stems from processing the calibration data.

### B.2.3. PRUNING DIFFERENT MODULES

In Section 3.4, we note that pruning different modules exerts heterogeneous effects on the overall performance of Mamba-based LLMs, with sensitivity varying markedly across modules. Specifically, pruning the in_proj module leads to a more pronounced performance drop than pruning other modules, and pruning the out_proj module similarly induces significant degradation, whereas remaining modules demonstrate higher resilience to parameter removal.

Within the Mamba architecture, the in_proj and out_proj modules serve as the principal input projection and output transformation layers, respectively, endowing them with high coupling and low redundancy that limit their prunability. Conversely, other modules are characterized by extensive overparameterization, enabling redundant representations of analogous functionalities and yielding comparatively low Hessian curvature across their parameters.

*Table 7.* Performance analysis results for pruning different modules. In each row, the Module column denotes the component being pruned, with 50% sparsity applied to the Mamba-370M model.

| Module | Wiki. ↓ | PTB ↓ | C4 ↓ | OBQA ↑ | PIQA ↑ | ARC-e ↑ | ARC-c ↑ | WinoG ↑ | Avg. ↑ |
|---|---|---|---|---|---|---|---|---|---|
| conv1d | 14.46 | 23.78 | 19.52 | 30.80 | 68.61 | 55.13 | 27.30 | 55.01 | 47.37 |
| in_proj | 16.28 | 27.23 | 22.68 | 30.40 | 66.43 | 51.56 | 26.88 | 55.25 | 46.10 |
| x_proj | 14.35 | 23.55 | 19.39 | 30.40 | 68.55 | 54.59 | 27.90 | 55.64 | 47.42 |
| dt_proj | 14.49 | 23.88 | 19.56 | 30.80 | 68.39 | 54.50 | 28.75 | 54.78 | 47.44 |
| out_proj | 15.19 | 25.45 | 21.47 | 31.00 | 66.87 | 54.08 | 27.56 | 56.12 | 47.13 |

### B.2.4. RESULTS OF SPEEDUP

As reported in Table 2, our pruning method yields consistent runtime acceleration across different model variants. In the recurrent mode, the state transition matrix $A_{log}$ is invoked repeatedly. The speedup becomes much more pronounced, reaching up to $2.23\times$ on Mamba-2.8B. This observation highlights that even though $A_{log}$ comprises a relatively small number of parameters, its repeated usage amplifies the benefits of pruning, especially in the absence of specialized CUDA kernels. Therefore, pruning directly translates into significant end-to-end runtime gains in practical inference scenarios.

All runtime experiments above were conducted on a single NVIDIA GeForce RTX 4090 GPU. Our primary latency

*Table 8.* Decode throughput under 50% structured pruning across different Mamba model sizes. Throughput is measured in tokens/s.

| Model | Dense (tokens/s) | Pruned (tokens/s) | Speedup |
|---|---|---|---|
| Mamba-370m | 56.986 | 85.483 | 1.500× |
| Mamba-790m | 52.952 | 78.039 | 1.474× |
| Mamba-1.4b | 49.751 | 73.658 | 1.481× |
| Mamba-2.8b | 34.744 | 67.598 | 1.946× |

*Table 9.* Speedup of selective scan under 50% structured pruning on the SSM module using the optimized `mamba-ssm` kernel.

| Model | Dense (ms) | Pruned (ms) | Speedup |
|---|---|---|---|
| Mamba-370m | 5.530 | 3.210 | 1.72× |
| Mamba-790m | 7.948 | 4.666 | 1.70× |
| Mamba-1.4b | 10.878 | 6.456 | 1.68× |
| Mamba-2.8b | 18.224 | 11.000 | 1.66× |

measurements are indeed based on real wall-clock timing by using `torch.profiler` to measure latency by directly benchmarking the pruned models, without relying on any additional acceleration frameworks.

We further evaluate the decoding speed before and after pruning, see Table 8. We use the official mamba-ssm kernel and measure latency with CUDA events, reporting the CUDA elapsed time speedup. Specifically, we use a prompt length of 2048, a decode length of 256, and a batch size of 1. We first perform 3 warm-up runs, and then run each experiment 10 times and report the mean latency.

Meanwhile, we specifically measure the latency of the selective scan operator before and after pruning, see Table 9. Using the official optimized kernel of Mamba-1 and the same other configuration, we apply structured pruning to the SSM module and perform inference with the official accelerated CUDA kernel provided by Mamba. The results below demonstrate that the current structured pruning method can provide effective real-world acceleration for the selective scan process.

Additionally, we do not compare the acceleration performance with other baselines, since they are all unstructured pruning methods. Without specialized acceleration kernels, their practical speedup cannot be directly observed. Therefore, we only report the acceleration results of the structured extension of our method without using any specialized acceleration kernels. This work only extends structured pruning to the SSM module. Therefore, we only report the speedup achieved by pruning the SSM module.

### B.2.5. RESULTS OF MEMORY FOOTPRINT

Table 10 reports the peak device memory before and after structured pruning across model scales and execution modes. Pruning compacts the SSM state and associated working sets, yielding large reductions in peak memory, e.g., from 12,897 MB to 6,872 MB on Mamba-2.8B, approaching a near-2× decrease, with consistent 30–45% savings on medium and large models in the recurrent settings. These results indicate that structured pruning in $A_{log}$ directly translates into materially lower memory headroom for recurrent inference, enabling larger batch sizes or longer contexts under fixed VRAM budgets.

### B.2.6. RESULTS OF ADDITIONAL SSM PRUNING EXPERIMENTS

To better demonstrate the robustness of our method, we additionally conduct more SSM pruning experiments.

We conduct experiments on the Mamba-2 architecture to validate the effectiveness of our method, see table 11. To further assess the scalability of our approach, we evaluate it on the Mamba2-8B model (Waleffe et al., 2024), a large-scale checkpoint trained with the Megatron framework. The results in table 12 confirm that our method remains effective at this larger model size.

We then ran additional experiments on a validation split of FineWeb (Penedo et al., 2024), treating it analogously to the first three perplexity-based benchmarks. Here, we used exactly the same evaluation pipeline as in our existing LM evaluations to ensure a fair comparison. On this large-scale web corpus, we consistently observe that *SparseSSM* yields lower perplexity than the training-free baselines at the same sparsity levels, see Table 13.

*Table 10.* Memory results of structured pruning across model sizes at 50% sparsity.

| Model | Memory (dense model) | Memory (50% sparsity) | Reduction (%) |
|---|---|---|---|
| Mamba-130m | 1195.1 | 1017.4 | 14.9 |
| Mamba-370m | 2354.0 | 1552.2 | 34.1 |
| Mamba-790m | 4433.0 | 2926.6 | 34.0 |
| Mamba-1.4b | 7099.6 | 4702.2 | 33.8 |
| Mamba-2.8b | 12897 | 6872.2 | 46.7 |

*Table 11.* Performance analysis for one-shot unstructured pruning of SSM modules in Mamba-2 models ($370M \sim 2.7B$) at $50\%$ sparsity. Here, $\downarrow$ lower metrics reflect better outcomes, and $\uparrow$ denotes higher metrics reflect better outcomes.

| Model | Method | Wiki. ↓ | PTB ↓ | C4 ↓ | OBQA ↑ | PIQA ↑ | ARC-e ↑ | ARC-c ↑ | WinoG ↑ | Avg. ↑ |
|---|---|---|---|---|---|---|---|---|---|---|
| | Dense | 14.17 | 22.01 | 19.07 | 32.40 | 69.26 | 54.84 | 26.71 | 55.41 | 47.72 |
| Mamba2-370M | SparseGPT (Frantar & Alistarh, 2023) | 15.42 | 24.33 | 20.90 | 31.20 | 69.26 | 53.79 | 26.54 | 54.38 | 47.03 |
| | *SparseSSM* | **14.89** | **23.48** | **19.92** | 31.60 | 68.93 | **54.55** | **27.13** | 54.38 | **47.32** |
| | Dense | 11.81 | 18.20 | 16.46 | 36.20 | 71.82 | 61.03 | 28.50 | 60.22 | 51.55 |
| Mamba2-780M | SparseGPT (Frantar & Alistarh, 2023) | 12.81 | 19.59 | 18.28 | **35.80** | 71.44 | 58.29 | 27.56 | 56.83 | 49.98 |
| | *SparseSSM* | **12.70** | **19.20** | **18.13** | 35.40 | **71.60** | **59.34** | **28.07** | **57.38** | **50.36** |
| | Dense | 10.42 | 15.86 | 14.78 | 37.80 | 73.72 | 64.14 | 33.11 | 61.01 | 53.96 |
| Mamba2-1.3B | SparseGPT (Frantar & Alistarh, 2023) | 12.01 | 17.72 | 16.97 | **36.20** | 73.50 | 63.05 | 32.17 | **61.17** | 53.22 |
| | *SparseSSM* | **10.88** | **16.55** | **15.72** | 35.80 | **73.56** | **63.80** | **32.68** | 60.93 | **53.35** |
| | Dense | 9.062 | 13.80 | 13.26 | 38.80 | 76.22 | 69.57 | 36.26 | 63.93 | 56.96 |
| Mamba2-2.7B | SparseGPT (Frantar & Alistarh, 2023) | 12.56 | 14.53 | 15.78 | 38.20 | **75.90** | 68.81 | **36.60** | 63.22 | 56.55 |
| | *SparseSSM* | **9.67** | **14.31** | **14.27** | **38.80** | **75.90** | **68.94** | 36.26 | **63.61** | **56.70** |

### B.2.7. RESULTS OF PRUNING OTHER SSM ARCHITECTURE

To test applicability beyond language, we apply *SparseSSM* to a vision SSM architecture, MambaVision (Hatamizadeh & Kautz, 2025), on the ImageNet-1K validation set. Using the same training-free pipeline, *SparseSSM* preserves top-1/top-5 accuracy almost perfectly and compares favorably to magnitude pruning at the same sparsity level (see Table 14), showing that our method extends naturally to a different modality and data distribution. Recent training-free approaches to improving LVM reliability, such as visual adversarial perturbations for mitigating object hallucinations (Zhang et al., 2025), further highlight that efficient and reliable visual backbones are both important for deployable multimodal systems. To probe hybrid architectures, we validate our method on a small open-source checkpoint, Jamba-tiny-dev, with results reported in Table 15. Recent multimodal reasoning models introduce dynamically evolving latent visual states for step-wise reasoning (Dong et al., 2025). This suggests that our time-aware saliency estimation may be useful for future state-evolving architectures beyond Mamba-style SSMs.

### B.2.8. DETAILED RESULTS OF ABLATION STUDY

We have explicitly compared power-law aggregation with uniform and exponential alternatives in our pruning experiments and found that the power-law choice consistently yields the best perplexity and zero-shot accuracy at matched sparsity levels. In our main experiments, we use an exponent of $\mu = 1.0$ for the power-law aggregation, which we found to be a simple and effective default. We also ran a sensitivity study around this choice, varying in a moderate range of $0.5 \sim 2.5$. The results in Table 16 show that *SparseSSM* is reasonably robust to the exact value of $\mu$. Perplexity and zero-shot accuracy change only marginally within this range, and the overall ranking against baselines remains unchanged. Small $\mu$ and large $\mu$ values lead to mildly worse performance.

We also provide an ablation study on different calibration data. Specifically, we include experiments using Wikitext-2, PTB,

*Table 12.* Performance analysis of the `mamba2-8b-3t-4k` checkpoint at $50\%$ sparsity.

| Method | Wiki. ↓ | PTB ↓ | C4 ↓ | FineWeb ↓ |
|---|---|---|---|---|
| Dense | 7.25 | 11.26 | 9.96 | 7.91 |
| *SparseSSM* | 7.52 | 11.85 | 10.30 | 8.22 |

*Table 13.* FineWeb perplexity under one-shot unstructured pruning of SSM modules at 50% sparsity. Lower perplexity (↓) indicates better performance.

| Method | FineWeb ppl. ↓ | | | |
|---|---|---|---|---|
| | Mamba-130M | Mamba-370M | Mamba-790M | Mamba-1.4B |
| Dense | 17.80 | 13.45 | 13.45 | 10.80 |
| SparseGPT (Frantar & Alistarh, 2023) | 6.5e6 | 4115 | 68.68 | 31.14 |
| *SparseSSM* | **18.22** | **14.14** | **12.12** | **11.94** |

*Table 14.* Performance analysis on the Imagenet-1K validation set of the MambaVision-B checkpoint at 50% sparsity. Higher accuracy (↑) and lower error (↓) indicate better performance.

| Method | Top-1 (%) ↑ | Top-1 error (%) ↓ | Top-5 (%) ↑ | Top-5 error (%) ↓ |
|---|---|---|---|---|
| Dense | 84.68 | 15.31 | 97.34 | 2.66 |
| Magnitude pruning | 84.36 | 15.64 | 97.12 | 2.88 |
| *SparseSSM* | **84.65** | **15.35** | **97.37** | **2.62** |

C4, and FineWeb as calibration datasets. As shown in the table 17, our method achieves consistently stable performance across different calibration datasets, demonstrating its robustness.

Pruning the FFN and SSM modules affects the final model performance differently. To investigate this effect, we separately evaluate pruning the SSM module, pruning the FFN, and pruning both modules jointly, see Table 18.

### B.2.9. RESULTS OF PRUNING SSM MODULE AT HIGH SPARSITY

We further compare our method against magnitude pruning (MP), Mamba-Shedder, and SparseGPT across a range of sparsity levels. The pruning results for these methods are reported on Mamba-130M, Mamba-370M, Mamba-790M, and Mamba-1.4B. We evaluate the perplexity of each pruned model on WikiText-2, PTB, and C4, and measure task accuracy on OpenBookQA, PIQA, ARC-Easy, ARC-Challenge, and Winogrande. As summarized in Table 19, 20, 21, 22, our approach consistently outperforms all baselines at every sparsity level, thereby demonstrating its robustness.

*Table 15.* Performance analysis on the Jamba-tiny-dev checkpoint. Higher scores (↑) indicate better performance.

| Method | OBQA ↑ | PIQA ↑ | ARC-e ↑ | ARC-c ↑ | WinoG ↑ | Avg. ↑ |
|---|---|---|---|---|---|---|
| Dense | 25.4 | **57.89** | **33.21** | 17.49 | 50.04 | 36.81 |
| *SparseSSM* | **25.8** | 57.40 | 32.62 | **18.01** | **50.36** | **36.84** |

*Table 16.* Ablation of temporal aggregation strategies and the exponent $p$ for power-law aggregation on Mamba-370M at 50% sparsity. Here, ↓ lower metrics reflect better outcomes, and ↑ denotes higher metrics reflect better outcomes.

| Method / $\mu$ | Wiki. ↓ | PTB ↓ | C4 ↓ | OBQA ↑ | PIQA ↑ | ARC-e ↑ | ARC-c ↑ | WinoG ↑ | Avg. ↑ |
|---|---|---|---|---|---|---|---|---|---|
| **(a) Ablation of aggregation methods** | | | | | | | | | |
| Linear aggregation | 15.28 | 24.93 | 20.47 | 29.20 | 68.61 | 53.75 | 26.45 | 54.70 | 46.54 |
| Exponential aggregation | 15.16 | 24.78 | 20.35 | 30.00 | **69.31** | **55.64** | 27.13 | 54.85 | 47.39 |
| **Power-law aggregation (ours)** | **15.04** | **24.63** | **20.13** | **30.20** | 68.34 | 55.26 | **27.90** | **55.64** | **47.47** |
| **(b) Effect of exponent $p$ in power-law aggregation** | | | | | | | | | |
| 0.5 | 15.12 | 24.78 | 20.31 | 29.60 | 68.66 | 55.01 | 26.71 | 55.01 | 47.00 |
| 1.0 | **15.04** | **24.63** | 20.13 | 30.20 | 68.34 | 55.26 | **27.90** | **55.64** | 47.47 |
| 1.5 | 15.06 | 24.64 | **20.11** | **30.80** | 69.04 | 55.77 | 27.22 | 54.93 | **47.55** |
| 2.0 | 15.17 | 24.81 | 20.28 | 30.20 | **69.15** | **56.27** | 27.22 | 54.70 | 47.51 |
| 2.5 | 15.30 | 25.15 | 20.45 | 30.00 | 68.66 | 55.81 | 26.96 | 54.93 | 47.27 |

*Table 17.* Results of unstructured pruning with 50% parameter sparsity under different calibration datasets. Here, ↓ lower metrics reflect better outcomes, and ↑ denotes higher metrics reflect better outcomes.

| Model | Calibration Data | Wiki. ↓ | PTB ↓ | C4 ↓ | OBQA ↑ | PIQA ↑ | ARC-e ↑ | ARC-c ↑ | WinoG ↑ | Avg. ↑ |
|---|---|---|---|---|---|---|---|---|---|---|
| Mamba-370m | Dense | 14.32 | 23.46 | 19.37 | 31.00 | 68.34 | 54.97 | 27.90 | 55.25 | 47.49 |
| | Wiki. | **15.04** | **24.63** | **20.13** | 30.20 | 68.34 | 55.26 | **27.90** | **55.64** | 47.47 |
| | PTB | 15.19 | 24.73 | 20.15 | 30.40 | 68.50 | 55.81 | 27.05 | 55.01 | 47.35 |
| | C4 | 15.18 | 24.76 | 20.25 | 30.20 | 68.88 | **55.85** | **27.90** | 54.70 | **47.51** |
| | Fineweb | 15.08 | 24.87 | 20.16 | **31.00** | **68.93** | 55.01 | 27.39 | 54.70 | 47.41 |

*Table 18.* Results of pruning 30% parameters by pruning the SSM module, the FFN, or both. Here, ↓ lower metrics reflect better outcomes, and ↑ denotes higher metrics reflect better outcomes.

| Model | Method | Wiki. ↓ | PTB ↓ | C4 ↓ | OBQA ↑ | PIQA ↑ | ARC-e ↑ | ARC-c ↑ | WinoG ↑ | Avg. ↑ |
|---|---|---|---|---|---|---|---|---|---|---|
| | Dense | 14.32 | 23.46 | 19.37 | 31.00 | 68.34 | 54.97 | 27.90 | 55.25 | 47.49 |
| Mamba-370M | Pruning SSM module | 14.38 | 23.56 | 19.42 | 30.20 | 69.10 | 55.13 | 28.33 | 55.41 | 47.63 |
| | Pruning FFN | 16.22 | 25.94 | 21.65 | 29.60 | 67.14 | 52.86 | 26.79 | 54.62 | 46.20 |
| | Pruning both | 16.37 | 26.56 | 21.90 | 30.00 | 67.14 | 52.23 | 27.82 | 54.78 | 46.39 |

*Table 19.* Performance analysis for one-shot unstructured pruning of SSM modules in Mamba models (130M ∼ 1.4B) at 40% sparsity. Here, ↓ lower metrics reflect better outcomes, and ↑ denotes higher metrics reflect better outcomes.

| Model | Method | Wiki. ↓ | PTB ↓ | C4 ↓ | OBQA ↑ | PIQA ↑ | ARC-e ↑ | ARC-c ↑ | WinoG ↑ | Avg. ↑ |
|---|---|---|---|---|---|---|---|---|---|---|
| | Dense | 20.60 | 32.75 | 25.66 | 28.60 | 63.28 | 48.02 | 24.40 | 52.5 | 43.36 |
| | MP (Han et al., 2015) | 218.7 | 304.86 | 107.77 | 28.20 | 60.72 | 40.57 | 23.29 | **51.85** | 40.93 |
| Mamba-130M | Mamba-Shedder (Munoz et al., 2025) | 275.3 | 506.6 | 222.8 | 25.00 | 55.11 | 34.89 | 22.10 | 49.72 | 37.37 |
| | SparseGPT (Frantar & Alistarh, 2023) | 165.0 | 211.3 | 87.22 | 28.80 | 59.96 | 40.66 | **24.74** | 50.43 | 40.92 |
| | *SparseSSM* | **20.84** | **33.04** | **25.85** | **30.40** | **63.49** | **45.79** | 24.49 | 50.75 | **42.98** |
| | Dense | 14.32 | 23.46 | 19.37 | 31.00 | 68.34 | 54.97 | 27.90 | 55.25 | 47.49 |
| | MP (Han et al., 2015) | 149.8 | 264.8 | 70.17 | 31.00 | 65.89 | 51.22 | 25.77 | 51.85 | 45.15 |
| Mamba-370M | Mamba-Shedder (Munoz et al., 2025) | 195.5 | 310.6 | 137.9 | 26.20 | 56.80 | 30.60 | 22.10 | 49.64 | 37.07 |
| | SparseGPT (Frantar & Alistarh, 2023) | 2.8e4 | 4.6e6 | 6367 | **31.80** | 65.89 | 50.84 | 26.54 | 53.04 | 45.62 |
| | *SparseSSM* | **14.52** | **23.85** | **19.58** | 30.40 | **69.10** | **55.35** | 27.90 | 54.78 | **47.50** |
| | Dense | 11.96 | 18.45 | 16.62 | 33.80 | 72.63 | 61.07 | 29.44 | 56.27 | 50.64 |
| | MP (Han et al., 2015) | 97.37 | 150.4 | 53.35 | 32.40 | 68.66 | 54.17 | 27.65 | 55.33 | 47.64 |
| Mamba-790M | Mamba-Shedder (Munoz et al., 2025) | 75.51 | 109.5 | 78.93 | 33.60 | 71.06 | 56.57 | 27.39 | 55.72 | 48.87 |
| | SparseGPT (Frantar & Alistarh, 2023) | 36.14 | 81.62 | 34.13 | 32.80 | 68.34 | 54.42 | 27.47 | 54.93 | 47.59 |
| | *SparseSSM* | **12.15** | **18.82** | **16.81** | **34.00** | **70.13** | **57.11** | **29.95** | **55.88** | **49.41** |
| | Dense | 10.75 | 17.05 | 15.17 | 36.40 | 73.88 | 65.57 | 32.85 | 61.17 | 53.98 |
| | MP (Han et al., 2015) | 49.99 | 84.70 | 34.14 | 34.60 | 70.35 | 59.68 | 27.82 | 56.04 | 49.70 |
| Mamba-1.4B | Mamba-Shedder (Munoz et al., 2025) | 120.6 | 179.5 | 109.7 | 26.40 | 60.45 | 39.86 | 22.95 | 52.41 | 40.41 |
| | SparseGPT (Frantar & Alistarh, 2023) | 32.39 | 49.87 | 28.86 | **36.20** | 72.36 | 61.49 | 31.48 | 57.30 | 51.77 |
| | *SparseSSM* | **11.24** | **17.94** | **15.78** | 35.40 | **73.01** | **63.30** | 32.08 | 57.38 | **52.24** |

*Table 20.* Performance analysis for one-shot unstructured pruning of SSM modules in Mamba models (130M ∼ 1.4B) at 60% sparsity. Here, ↓ lower metrics reflect better outcomes, and ↑ denotes higher metrics reflect better outcomes.

| Model | Method | Wiki. ↓ | PTB ↓ | C4 ↓ | OBQA ↑ | PIQA ↑ | ARC-e ↑ | ARC-c ↑ | WinoG ↑ | Avg. ↑ |
|---|---|---|---|---|---|---|---|---|---|---|
| | Dense | 20.60 | 32.75 | 25.66 | 28.60 | 63.28 | 48.02 | 24.40 | 52.50 | 43.36 |
| | MP (Han et al., 2015) | 1034 | 1605 | 351.7 | 26.00 | 55.55 | 33.42 | 22.10 | 49.96 | 37.41 |
| Mamba-130M | Mamba-Shedder (Munoz et al., 2025) | 3219 | 4998 | 1503 | 25.80 | 54.46 | 29.00 | 23.72 | 50.04 | 36.60 |
| | SparseGPT (Frantar & Alistarh, 2023) | 5.0e4 | 1.4e4 | 2.4e4 | 26.20 | 52.45 | 26.85 | 23.55 | 49.80 | 35.77 |
| | *SparseSSM* | **22.31** | **35.78** | **27.45** | **30.20** | **63.38** | **45.92** | **24.40** | **52.17** | **43.21** |
| | Dense | 14.32 | 23.46 | 19.37 | 31.00 | 68.34 | 54.97 | 27.90 | 55.25 | 47.49 |
| | MP (Han et al., 2015) | 386.2 | 747.6 | 141.6 | 26.40 | 58.05 | 38.64 | 21.59 | 49.64 | 38.86 |
| Mamba-370M | Mamba-Shedder (Munoz et al., 2025) | 463.3 | 561.6 | 307.0 | 25.00 | 54.03 | 28.91 | 23.63 | 49.72 | 36.26 |
| | SparseGPT (Frantar & Alistarh, 2023) | 360.2 | 1455 | 324.7 | 30.00 | 58.87 | 40.07 | 23.89 | 53.28 | 41.22 |
| | *SparseSSM* | **16.44** | **26.47** | **21.62** | 29.60 | **68.06** | **55.05** | **26.54** | **53.99** | **46.65** |
| | Dense | 11.96 | 18.45 | 16.62 | 33.80 | 72.63 | 61.07 | 29.44 | 56.27 | 50.64 |
| | MP (Han et al., 2015) | 255.6 | 502.5 | 108.4 | 28.40 | 60.61 | 41.92 | 23.29 | 51.85 | 41.22 |
| Mamba-790M | Mamba-Shedder (Munoz et al., 2025) | 353.5 | 358.3 | 283.5 | 26.60 | 54.95 | 32.58 | 23.04 | 49.96 | 37.43 |
| | SparseGPT (Frantar & Alistarh, 2023) | 1033 | 3630 | 897.5 | 31.40 | 65.40 | 51.26 | 24.74 | 53.67 | 45.29 |
| | *SparseSSM* | **13.24** | **21.21** | **17.96** | **31.80** | **70.29** | **55.01** | **28.67** | **55.41** | **48.24** |
| | Dense | 10.75 | 17.05 | 15.17 | 36.40 | 73.88 | 65.57 | 32.85 | 61.17 | 53.98 |
| | MP (Han et al., 2015) | 150.9 | 322.3 | 67.64 | 30.20 | 62.73 | 47.47 | 25.43 | 50.99 | 43.36 |
| Mamba-1.4B | Mamba-Shedder (Munoz et al., 2025) | 370.4 | 481.4 | 281.4 | 26.80 | 55.55 | 33.67 | 23.29 | 50.67 | 38.00 |
| | SparseGPT (Frantar & Alistarh, 2023) | 110.3 | 209.2 | 70.36 | **34.60** | 69.91 | 58.59 | 27.99 | 53.75 | 48.97 |
| | *SparseSSM* | **13.51** | **21.10** | **18.34** | 31.60 | **70.95** | **58.80** | **29.44** | **54.70** | **49.10** |

*Table 21.* Performance analysis for one-shot unstructured pruning of SSM modules in Mamba models (130M ∼ 1.4B) at 70% sparsity. Here, ↓ lower metrics reflect better outcomes, and ↑ denotes higher metrics reflect better outcomes.

| Model | Method | Wiki. ↓ | PTB ↓ | C4 ↓ | OBQA ↑ | PIQA ↑ | ARC-e ↑ | ARC-c ↑ | WinoG ↑ | Avg. ↑ |
|---|---|---|---|---|---|---|---|---|---|---|
| | Dense | 20.60 | 32.75 | 25.66 | 28.60 | 63.28 | 48.02 | 24.40 | 52.50 | 43.36 |
| | MP (Han et al., 2015) | 1248 | 1802 | 407.0 | 24.80 | 54.13 | 30.68 | 24.32 | **52.49** | 37.28 |
| Mamba-130M | Mamba-Shedder (Munoz et al., 2025) | 5845 | 1.2e4 | 3775 | 26.80 | 51.85 | 26.56 | **24.57** | 50.67 | 36.09 |
| | SparseGPT (Frantar & Alistarh, 2023) | 1.1e5 | 6.7e4 | 1.8e5 | 24.20 | 51.47 | 25.59 | 24.40 | 50.36 | 35.20 |
| | *SparseSSM* | **25.71** | **40.20** | **30.75** | **29.00** | **62.40** | **44.61** | 24.15 | 52.33 | **42.50** |
| | Dense | 14.32 | 23.46 | 19.37 | 31.00 | 68.34 | 54.97 | 27.90 | 55.25 | 47.49 |
| | MP (Han et al., 2015) | 497.3 | 925.2 | 174.4 | 25.60 | 56.91 | 36.70 | 20.05 | 51.30 | 38.11 |
| Mamba-370M | Mamba-Shedder (Munoz et al., 2025) | 1029 | 933.0 | 625.7 | 26.80 | 52.67 | 27.95 | 23.63 | 50.28 | 36.26 |
| | SparseGPT (Frantar & Alistarh, 2023) | 7.8e4 | 5.5e4 | 7.3e4 | 27.80 | 59.30 | 39.23 | 22.61 | 50.36 | 39.86 |
| | *SparseSSM* | **19.74** | **30.88** | **25.18** | **28.40** | **67.30** | **52.61** | **24.57** | **52.25** | **45.03** |
| | Dense | 11.96 | 18.45 | 16.62 | 33.80 | 72.63 | 61.07 | 29.44 | 56.27 | 50.64 |
| | MP (Han et al., 2015) | 341.2 | 655.9 | 139.6 | 27.00 | 57.83 | 38.85 | 24.57 | 51.22 | 39.90 |
| Mamba-790M | Mamba-Shedder (Munoz et al., 2025) | 353.5 | 358.3 | 283.5 | 26.60 | 54.95 | 32.58 | 23.04 | 49.96 | 37.43 |
| | SparseGPT (Frantar & Alistarh, 2023) | 1.9e5 | 2.4e7 | 2.7e5 | 27.60 | 61.32 | 39.10 | 24.49 | 52.96 | 41.09 |
| | *SparseSSM* | **14.92** | **24.49** | **19.92** | **31.20** | **69.26** | **55.05** | **28.07** | **54.85** | **47.69** |
| | Dense | 10.75 | 17.05 | 15.17 | 36.40 | 73.88 | 65.57 | 32.85 | 61.17 | 53.98 |
| | MP (Han et al., 2015) | 180.8 | 378.5 | 80.16 | 28.80 | 59.58 | 41.67 | 23.55 | 51.07 | 40.93 |
| Mamba-1.4B | Mamba-Shedder (Munoz et al., 2025) | 805.1 | 796.6 | 541.7 | 25.40 | 54.08 | 29.50 | 24.06 | 49.09 | 36.43 |
| | SparseGPT (Frantar & Alistarh, 2023) | 452.5 | 602.9 | 253.7 | **31.20** | 66.27 | 54.00 | 24.40 | 50.36 | 45.24 |
| | *SparseSSM* | **17.91** | **28.14** | **23.72** | 29.20 | **68.72** | **54.04** | **26.96** | **53.75** | **46.53** |

*Table 22.* Performance analysis for one-shot unstructured pruning of SSM modules in Mamba models (130M ∼ 1.4B) at 80% sparsity. Here, ↓ lower metrics reflect better outcomes, and ↑ denotes higher metrics reflect better outcomes.

| Model | Method | Wiki. ↓ | PTB ↓ | C4 ↓ | OBQA ↑ | PIQA ↑ | ARC-e ↑ | ARC-c ↑ | WinoG ↑ | Avg. ↑ |
|---|---|---|---|---|---|---|---|---|---|---|
| | Dense | 20.60 | 32.75 | 25.66 | 28.60 | 63.28 | 48.02 | 24.40 | 52.50 | 43.36 |
| | MP (Han et al., 2015) | 1297 | 1870 | 420.4 | 24.00 | 52.18 | 31.31 | **24.32** | 50.51 | 36.46 |
| Mamba-130M | Mamba-Shedder (Munoz et al., 2025) | 2.6e4 | 5.9e4 | 2.2e4 | 26.20 | 51.69 | 28.03 | 23.89 | 52.01 | 36.36 |
| | SparseGPT (Frantar & Alistarh, 2023) | 2.6e21 | 5.7e22 | 2.7e23 | 24.80 | 55.98 | 30.60 | 23.38 | 51.30 | 37.21 |
| | *SparseSSM* | **38.61** | **55.81** | **41.88** | **27.80** | **61.32** | **42.59** | 23.04 | **52.09** | **41.37** |
| | Dense | 14.32 | 23.46 | 19.37 | 31.00 | 68.34 | 54.97 | 27.90 | 55.25 | 47.49 |
| | MP (Han et al., 2015) | 538.2 | 983.0 | 191.0 | 25.20 | 53.16 | 31.99 | 22.61 | 49.49 | 36.49 |
| Mamba-370M | Mamba-Shedder (Munoz et al., 2025) | 3191 | 933.0 | 1848 | **27.80** | 52.34 | 26.52 | **24.06** | 51.14 | 36.37 |
| | SparseGPT (Frantar & Alistarh, 2023) | 1.1e5 | 1.2e5 | 1.0e5 | 27.40 | 56.26 | 34.93 | 23.38 | **53.83** | 39.16 |
| | *SparseSSM* | **28.99** | **40.74** | **34.45** | 27.40 | **65.72** | **49.12** | 23.72 | 51.93 | **43.58** |
| | Dense | 11.96 | 18.45 | 16.62 | 33.80 | 72.63 | 61.07 | 29.44 | 56.27 | 50.64 |
| | MP (Han et al., 2015) | 402.7 | 738.5 | 160.1 | 25.80 | 56.80 | 36.66 | 22.70 | 49.41 | 38.27 |
| Mamba-790M | Mamba-Shedder (Munoz et al., 2025) | 1891 | 2121 | 1277 | 25.40 | 51.69 | 28.28 | 24.40 | 48.15 | 35.58 |
| | SparseGPT (Frantar & Alistarh, 2023) | 1.7e8 | 4.3e8 | 2.1e8 | 27.40 | 56.80 | 36.70 | 23.12 | 50.91 | 38.99 |
| | *SparseSSM* | **19.92** | **31.09** | **25.19** | 27.40 | **67.14** | **54.29** | **27.73** | **52.49** | **45.81** |
| | Dense | 10.75 | 17.05 | 15.17 | 36.40 | 73.88 | 65.57 | 32.85 | 61.17 | 53.98 |
| | MP (Han et al., 2015) | 227.4 | 438.6 | 101.4 | 25.00 | 56.04 | 34.81 | 22.61 | **53.12** | 38.31 |
| Mamba-1.4B | Mamba-Shedder (Munoz et al., 2025) | 2260 | 2236 | 1405 | 26.80 | 51.20 | 28.87 | **27.13** | 51.14 | 37.03 |
| | SparseGPT (Frantar & Alistarh, 2023) | 5.7e11 | 2.6e13 | 3.1e14 | **28.20** | 59.36 | 43.22 | 23.38 | 48.93 | 40.62 |
| | *SparseSSM* | **29.79** | **58.01** | **36.38** | 26.60 | **64.42** | **48.32** | 24.15 | 49.72 | **42.64** |

