# OpenReview forum: "SparseSSM: Efficient Selective Structured State Space Models Can Be Pruned in One-Shot"
_ICML.cc/2026/Conference — ICML 2026 regular_

### Official Review · Reviewer_nMJV · 2026-02-20

**Soundness:** 3
**Presentation:** 3
**Significance:** 3
**Originality:** 3
**Overall Recommendation:** 5
**Confidence:** 3

**Summary:**

This paper proposed SpareSSM, a trianing-free, one-shot pruning framework developed for Mamba style selective SSM models.

The proposed method different from previous method by they can handle the time-shared and discretized state-transition parameter which serves as the core of Mamba's SSM module.

Empriically, the author shown that their method can prune 50% of the SSM transition parameter without finetuning and preserve the perpelxity and zero-shot accuracy.

**Compliance With Llm Reviewing Policy:**

Affirmed.

**Final Justification:**

My main concern has been solved by the author. So I decide to raise my score.

**Key Questions For Authors:**

Please refer to the weakness section above.

**Limitations:**

yes

**Strengths And Weaknesses:**

**Strengths**

- The paper identifies a concrete technical obstacle in transferring existing one-shot pruning methods to Mamba and proposes a pruning criterion and aggregation strategy explicitly designed around that structure.

- The pruning rule for the SSM transition parameters is computationally efficient: it uses hidden-state statistics gathered during forward passes and avoids full Hessian construction, consistent with practical OBS approximations.
- The empirical section is relatively thorough in scope: it includes multiple model scales (Mamba-1 and Mamba-2 variants), multiple LM perplexity benchmarks, several common zero-shot QA/reasoning tasks, plus ablations on aggregation and calibration size.
- Pruning/state compression for Mamba-family models is practically relevant: even if inference is asymptotically linear, deployment is still constrained by parameter count and repeated recurrent computation, so effective one-shot pruning could reduce cost in settings where fine-tuning is expensive.
- The work also contributes practical insights about heterogeneous pruneability inside Mamba blocks (which modules are most sensitive).



**Weakness**

1. There are places where the notation and derivation around discretization and the A/Alog parameterization are hard to reconcile cleanly across the main text and appendix. Can you clearly define the exact parameterization used in code for $A$, $A_{\log}$, and $\Delta A$, and ensure that Eq. (2), Theorem 3.1 (Eq. (6)), and the derivation in Appendix A to be aligned?
2.  In Table 2 (speedups), is the reported acceleration obtained from structured pruning with dimension reduction, or from unstructured sparsity alone? Please specify inference mode (recurrent vs. parallel), sequence length, batch size, and whether any sparse kernels are used.
3. How much total parameter reduction and latency reduction do you achieve when pruning the *entire* model at higher sparsities (e.g., 50% overall), and what is the best achievable trade-off without fine-tuning?
4. How sensitive are results to the choice of calibration data (WikiText-2 vs. other corpora) beyond the sample-size ablation?

---

> ### Author Rebuttal · Authors · 2026-03-31
>
> We sincerely thank reviewer `nMJV` for the careful reading, positive assessment, and constructive questions.
>
> ---
>
> > ***(W1):** Notation around discretization and (A/A_{\log}) is hard to reconcile across Eq. (2), the theorem, and Appendix A. Please define the exact parameterization used in the code and align Eq. (2), the theorem, and the appendix.*
>
> **Reply:** We thank the reviewer for pointing this out. We will align the notation across Eq. (2), the theorem, and Appendix A, and revise the proof to remove the current inconsistencies. Specifically, in our code implementation:
> - $ A_{log} $ is the stored trainable parameter, as defined in the official Mamba checkpoint.
> - $ A $ is the corresponding continuous-time transition parameter, implemented in code as `A = -torch.exp(self.A_log.float())`.
> - $ \Delta A $ is the discretized transition multiplier that is actually used in the recurrence, implemented in code as `deltaA = torch.exp(einsum(delta, A, "b l d_in, d_in n -> b l d_in n"))`.
>
> We will further unify the notation throughout the theorem statement and the appendix proof to make the implementation-to-theory correspondence fully explicit.
>
> ---
>
> > ***(W2):** Table 2 speedups: is the reported acceleration obtained from structured pruning with dimension reduction? Specify inference mode, sequence length, batch size, and sparse kernels.*
>
> **Reply:** Our method is primarily an unstructured pruning approach, while for pruning the SSM module we further **extend it to a structured pruning setting**. In the experiments reported in Table 2 in the paper, the speedup is obtained by structured pruning with dimension reduction on the SSM module. Specifically, we prune along the second axis of $A_{\log}$ and resize the $x_\text{proj}$ output dimension accordingly. For speed measurement, we use `torch.profiler` to measure end-to-end wall-clock speedup, and report the mean and standard deviation over 5 runs.
>
> We perform inference in recurrent mode with sequence length 2048 and batch size 1, without using any acceleration kernels or sparse kernels; the reported speedup is obtained directly under the native implementation.
>
> ---
>
> > ***(W3):** Total parameter/latency reduction when pruning the whole model at high sparsities; best trade-off without fine-tuning.*
>
> **Reply:** When pruning the entire model, at 50% sparsity the total number of parameters in Mamba-370M is reduced by 43.02% (the `lm_head` and one-dimensional vectors in the model are not pruned).
> - **As for the trade-off between latency and sparsity**, when pruning the entire model our method remains an unstructured pruning approach. Extending it to structured pruning over the full model while maintaining near-lossless performance is an important direction for future work, which we discuss in Section 5.
> - **As for the trade-off between performance and sparsity**, our method incurs almost no performance degradation at 20% sparsity. At higher sparsity levels, however, Mamba is inherently less tolerant to pruning than Transformer-based architectures, since its information representation is more compact by design.
>
> Table 1: Results at different sparsity levels when pruning the entire model
> | Model | Sparsity | Wiki. ↓ | PTB ↓ | C4 ↓ | OBQA ↑ | PIQA ↑ | ARC-e ↑ | ARC-c ↑ | WinoG ↑ | Avg. ↑ |
> | --- | --- | --- | --- | --- | --- | --- | --- | --- | --- | --- |
> | Mamba-370m | Dense | 14.32  | 23.46  | 19.37  | 31.00  | 68.34  | 54.97  | 27.90  | 55.25  | 47.49 |
> |  | 15% | 14.48 | 23.70 | 19.58 | 31.20 | 69.10 | 54.71 | 28.24 | 55.88 | 47.83 |
> |  | 20% | 14.79 | 24.27 | 19.95 | 31.00 | 68.01 | 53.54 | 27.90 | 55.49 | 47.19 |
>
> ---
>
> > ***(W4):** Sensitivity to calibration corpus choice (WikiText-2 vs others), beyond sample-size ablation.*
>
> **Reply:** We have added an ablation study on different calibration data. Specifically, we include experiments using Wikitext-2, PTB, C4, and FineWeb as calibration datasets. The results are shown below.
>
> Table 2: Results of unstructured pruning with 50% parameter sparsity under different calibration data.
> | Model | Calibration Data | Wiki. ↓ | PTB ↓ | C4 ↓ | OBQA ↑ | PIQA ↑ | ARC-e ↑ | ARC-c ↑ | WinoG ↑ | Avg. ↑ |
> | --- | --- | --- | --- | --- | --- | --- | --- | --- | --- | --- |
> | Mamba-370m | Dense | 14.32  | 23.46  | 19.37  | 31.00  | 68.34  | 54.97  | 27.90  | 55.25  | 47.49 |
> |  | Wikitext-2 | 15.04  | 24.63  | 20.13  | 30.20  | 68.34  | 55.26  | 27.90  | 55.64  | 47.47 |
> |  | PTB | 15.19 | 24.73 | 20.15 | 30.40 | 68.50 | 55.81 | 27.05 | 55.01 | 47.35 |
> |  | C4 | 15.18 | 24.76 | 20.25 | 30.20 | 68.88 | 55.85 | 27.90 | 54.70 | 47.51 |
> |  | FineWeb | 15.08 | 24.87 | 20.16 | 31.00 | 68.93 | 55.01 | 27.39 | 54.70 | 47.41 |
>
> As shown in the table, our method achieves **consistently stable performance** across different calibration datasets, demonstrating its robustness.
>
> ---
>
> We thank the reviewer again for these thoughtful comments. We would be happy to clarify any remaining questions during the next period.

---

> > ### Author Rebuttal · Reviewer_nMJV · 2026-04-05
> >
> > Thank you for the detailed rebuttal. My main concerns have been addressed.
> >
> > In particular, the rebuttal now clearly explains the parameterization and implementation correspondence among $A_{\log}$, $A$, and $\Delta A$, which resolves my earlier confusion about the consistency between Eq. (2), Theorem 3.1, Appendix A, and the actual code path.
> >
> > The speedup setting is also much clearer now: the reported gains come from structured pruning with dimension reduction on the SSM module, evaluated in recurrent mode with sequence length 2048 and batch size 1, without relying on sparse kernels. This clarification is important for properly interpreting Table 2.
> >
> > I also appreciate the added results on higher-sparsity trade-offs and the calibration-data ablation across WikiText-2, PTB, C4, and FineWeb, which strengthen the empirical picture and suggest that the method is reasonably robust.
> >
> > Overall, the rebuttal substantially improves the clarity of the paper and resolves the main issues I raised.

---

> > > ### Author Response · Authors · 2026-04-08
> > >
> > > Dear Reviewer `nMJV`,
> > >
> > > Thank you very much for revisiting our work and for deciding to raise the score.
> > >
> > > It is very encouraging to know that our clarifications and added results **improved the clarity of the paper and helped resolve the main issues you raised**. We also sincerely appreciate the time and care you devoted to reading our rebuttal and for your constructive feedback throughout the process.
> > >
> > > Best regards,
> > >
> > > The Authors of Submission 24916

---

### Official Review · Reviewer_U5B8 · 2026-02-20

**Soundness:** 2
**Presentation:** 3
**Significance:** 3
**Originality:** 2
**Overall Recommendation:** 4
**Confidence:** 4

**Summary:**

In this work, the authors propose a dedicated training-free pruning strategy for state space models (SSMs) and provide extensive evaluation of its impact on accuracy at different sparsity levels.

In details, the paper first introduces and adapts the standard optimal brain surgeon (OBS) pruning techniques for the core block of SSMs, and especially the state transition matrix A. They propose a new pruning protocol considering the time-sharing of this parameter using some effective hessian score computation and a time weighted aggregation.

This method, along with standard OBS pruning on the whole model, is then applied to the Mamba model and variations of it to evaluate its impact on performance at different sparsity levels and model scales, demonstrating a clear advantage over previous methods for accuracy.

Finally the authors run their pruning in a structured fashion, reporting the corresponding recurrent-mode speed-up, and performance evaluation on the 370M Mamba model.

**Compliance With Llm Reviewing Policy:**

Affirmed.

**Final Justification:**

The rebuttal addressed my concerns on the performance degradation due to structured sparsity (vs unstructured) and the effective speed-up gains achieved by the models.

Especially "Table 1: Results of structured pruning on the SSM module across different model scales" from the author's first rebuttal show the limited performance drop with structured sparsity, also at scale, and especially beating the previous SOTA (MP).

And "Table 2: Speedup of 50% structured pruning on the SSM module using the optimized mamba-ssm kernel"in the second author rebuttal shows the effective speed-ups with structured sparsity using the optimized mamba-ssm kernel.

I therefore increased my note from Weak Reject to Weak Accept.
I am still at Weak Accept vs Full Accept as there are still a few missing aspects/limitations:
* The main focus is on Mamba1 instead of the most widely adopted and relevant Mamba2
* Missing Pareto curves or tables equivalent to Table 2 in the paper but with the updated mamba-ssm kernel based speeds to check if it is in fact a preferred choice to take a larger model and prune it vs a smaller dense one for accuracy vs speed.

**Key Questions For Authors:**

1. How is the speed-up exactly measured? Is the model run fully recurrently on a defined sequence or is only the SSM core run recurrently while FFNs are applied in parallel over time?
2. What is the specific configuration employed in Table 2? Which sparsity level and which N:M  sparsity structure pattern?
3. Is there a novelty in the sensibility-aware pruning introduced in section 3.4 with respect to (Shao et al., 2024)? If yes it should be highlighted more and else the reuse of their method should be stated more clearly. As it is, it sounds like Shao et al. only proposed the sensitivity analysis but not the related strategy.
4. How are the speed-up gains from the state transition matrix A pruning transferring to Mamba2 where there is more parameter-sharing less independent parameters to prune?
5. Are you planning to release the code in case of acceptance?

**Limitations:**

yes

**Strengths And Weaknesses:**

**Strengths:**

1. The study provides extensive accuracy results on a wide range of model scales going up to 2.8B models, along with tackling different model variations (Mamba2, MambaVision, Jamba).
2. The proposed adapted OBS pruning method for the SSM core block is new, sound and justified theoretically, and show its benefits with the ablation study and the comparison with the SparseGPT method.
3. The achieved speed-up with structured sparsity is significant, especially on large models.
4. Overall the paper is well written and easy to follow.


**Weaknesses:**

1. The effective speed gains, which are the main objective of pruning methods, are only reported in the all-weight, structured pruning case, where the performance degradation is studied only partially. In fact, the Table 2 speed-ups are only correlated with the Table 3 and Table 4 performances on the 370M model and only compared to the oldest least efficient MP method. As it is, this questions the significance of the obtained results, as one cannot see if the most favorable speed-ups on large model are reached with decent performance or not. One would like to see the extensive performance analysis of Table 1, associated with the speed-up metrics.
2. The speed-up measurements are not standard for LLMs and also not defined clearly. The authors measure the speed through one forward pass, but do not indicate any selected dataset or sequence length. Considering the model is tested in recurrent mode, the measure should be done on a generative task with measuring latency and throughput in the prefill (parallel scan) and then decoding (recurrent) mode when new tokens get generated.
3. Directly related to point 1, speed-up gains are not reported for the other methods, which is unfair considering other structured methods like Mamba-shredder might have a harder time reaching high sparsity at iso-performance but can offer higher speed-ups at iso-sparsity due to the very dedicated structure of their sparsity.
4. In the same vein, this work is missing the key ablation study of speed-up achieved with the specific proposed pruning of the state transition matrix A versus pruning only standard weight matrixes. This leaves this claim unfounded: *“Although the number of parameters in Alog is relatively small, its contribution to acceleration is substantial because it is repeatedly utilized.”*
5. Tables and Figures titles are missing elements for direct understanding of the corresponding protocol and sometimes elements that are not even in the text. For instance Table 2 doesn't indicate the corresponding level of sparsity (guessing 50% from the text but it's not stated clearly) or that this is for structured sparsity and Table 1 doesn't indicate that the pruning study is only on the A matrix. This hinders clarity and could lead readers to interpret wrong results.
6. The performance degradation in Table 1 is increasing with model size, leading to think that the method might be less robust for large scale models. This goes for instance the opposite of observed dynamics in quantized models where scale makes models more resilient to quantization. A clear representation of effective speed-up per sparsity level would also help as larger models might need less sparsity for the same speed-up and thus can maintain higher performances.
7. One can regret that the more extensive study is done on the Mamba 1 instead of Mamba 2 model, which is the common one for modern LLM applications.

---

> ### Author Rebuttal · Authors · 2026-03-31
>
> We sincerely thank reviewer `U5B8` for the detailed comments. Below we respond point‑by‑point.
>
> ---
>
> > ***(W1/Q2):** Performance degradation is only partially analyzed; want performance analysis associated with speed.*
>
> **Reply:** Table 2 in the paper reports the results of extending our method to structured pruning on the SSM module, where the corresponding dimensions are reduced.
>
> For performance analysis associated with speed, we have added **a more complete set of experiments on structured pruning**, including both performance and speedup results.
>
> Table 1: Results of structured pruning on the SSM module across different model scales
> | Model | Sparsity | Speedup | Wiki. ↓ | PTB ↓ | C4 ↓ | Avg. zero-shot accuracy↑ |
> | --- | --- | --- | --- | --- | --- | --- |
> | Mamba-130m | Dense | / | 20.60 | 32.75 | 25.66 | 43.36 |
> |  | 50% | 1.08x | 35.16 | 55.12 | 33.52 | 42.08 |
> | Mamba-370m | Dense | / | 14.32  | 23.46  | 19.37  | 47.49 |
> |  | 50% | 1.66x | 18.13 | 28.82 | 22.65 | 46.31 |
> | Mamba-790m | Dense | / | 11.96 | 18.45 | 16.62 | 50.64 |
> |  | 50% | 1.70x | 16.69 | 24.77 | 20.00 | 47.21 |
> | Mamba-1.4b | Dense | / | 10.75 | 17.05 | 15.17 | 53.98 |
> |  | 50% | 1.66x | 14.94 | 21.96 | 18.17 | 50.13 |
> | Mamba-2.8b | Dense | / | 9.45 | 14.79 | 13.61 | 56.91 |
> |  | 50% | 2.23x | 12.20 | 18.46 | 15.84 | 51.51 |
>
> ---
>
> > ***(W2/Q1):** Speed-up measurements are not standard; should measure prefill and decoding in generative settings.*
>
> **Reply:** For speed measurement, we use inputs with sequence length 2048 and batch size 1. We use `torch.profiler` to measure end-to-end wall-clock latency, and report the mean and standard deviation over 5 runs.
>
> We report full-sequence forward latency as a proxy for end-to-end acceleration after structured pruning. We will further validate the results by testing prefill and decoding latency.
>
> ---
>
> > ***(W3):** Speed-up gains not reported for other methods*
>
> **Reply:** For other methods such as SparseGPT, since they are unstructured pruning methods, practical wall-clock speedup typically requires specially designed or dedicated sparse kernels. For Mamba-Shedder, the realized speedup needs modification on the mamba model, which makes a direct comparison less straightforward. For this reason, in the current version we mainly report the performance–sparsity trade-off for such methods.
>
> ---
>
> > ***(W4):** Missing ablation: speed-up from pruning (A) vs pruning other weights.*
>
> **Reply:** Our method is inherently an unstructured pruning approach. We extend it to structured pruning on the SSM module, rather than on the FFN part. Below, we provide the forward-pass latency of different modules to support the claim.
>
> Table 2: CUDA forward-pass time of different modules in Mamba-370m
> | **Module** | **inference time (ms)** |
> | --- | --- |
> | selective scan | 535.1 |
> | FFN | 12.03 |
>
> ---
>
> > ***(W5):** Table and figure titles are missing key protocol details.*
>
> **Reply:** We will revise the table and figure titles as follows:
> - For Table 1, we will state that the pruning study only affects the matrix $A$ in the SSM module.
> - For Table 2, we will revise the title to clearly indicate that it reports the results of structured pruning on the SSM module at 50% sparsity.
>
> ---
>
> > ***(W6):** Performance degradation increases with model size.*
>
> **Reply:** This phenomenon may be related to the fact that information is more compactly represented in Mamba models. In particular, for the 2.8B variant, the model appears to be more information-dense and is therefore less tolerant to pruning.
>
> However, for perplexity, the increase remains below 2.2 across all model scales, **which demonstrates the robustness of our approach**. We will further supplement the results at different sparsity levels in the revision.
>
> ---
>
> > ***(W7/Q4):** More extensive study is on Mamba-1 rather than Mamba-2; how do gains transfer to Mamba-2?*
>
> **Reply:** The submission already includes evaluations on Mamba-2 and its variants, and we plan to further conduct a more comprehensive study on Mamba-2 in future work.
>
> The SSD module in Mamba-2 still contains the computationally intensive state-space computation. Therefore, once structured pruning is further extended to Mamba-2, we expect that end-to-end speedup can still be achieved in this setting.
>
> ---
>
> > ***(Q3):** Novelty of sensitivity-aware pruning vs Shao et al. 2024?*
>
> **Reply:** The novelty of our sensitivity-aware pruning lies primarily in our **analysis of the sensitivity of different components in the Mamba architecture using both the Hessian trace and the reconstruction error**, as illustrated in Fig. 3. In contrast, from Shao et al. (2024), we mainly adopt the definition of the sparsity ratio.
>
> ---
>
> > ***(Q5):** Code release?*
>
> **Reply:** We will release the full codebase, including model pruning and evaluation, immediately upon acceptance.
>
> ---
>
> We thank the reviewer again for these thoughtful comments. We are happy to continue discussing any remaining questions in detail.

---

> > ### Author Rebuttal · Reviewer_U5B8 · 2026-04-04
> >
> > I thank the authors for their detailed rebuttal answers.
> >
> > The extra presented table allows to see the most relevant model performances, with a maintained gain with respect to the previous MP structured pruning method (mostly in perplexity).
> >
> > However, the reported inference times make me realize that the naive implementation of the SSM in recurrent mode is highly inefficient. Although this mode might be used on neuromorphic hardware, the fact measurements are made on a GPU with its central memory transfer bottleneck, render the efficiency gains highly overestimate wrt what one would obtain with any practical implementation.
> > Ideally this should be solved with implementing a dedicated kernel for sparse recurrent inference or having a neuromorphic implementation.
> >
> > I therefore prefer to maintain my current score for now.

---

> > > ### Author Response · Authors · 2026-04-08
> > >
> > > Dear Reviewer `U5B8`,
> > >
> > > We thank you for the careful follow-up and for acknowledging that the additional table makes the quality comparison clearer. We would like to further clarify the speedup issue.
> > >
> > > ---
> > >
> > > > *Reported speedups may be overestimated, because they are measured with a naive SSM implementation on GPU, rather than with a practical optimized kernel or hardware-aware implementation.*
> > >
> > > Our method is mainly an unstructured pruning approach for the Mamba architecture. On top of this, we have already extended our method to structured pruning of the SSM module in the current paper. To further demonstrate the practical speedup **under a standard, optimized setting**, we have now added new acceleration results.
> > >
> > > We have followed your suggestion and **evaluated the model’s prefill and decode latency on a generative task**. We use the official mamba-ssm kernel, and measure latency with CUDA events, reporting the CUDA elapsed time speedup. Specifically, we use a prompt length of 2048, a decode length of 256, and a batch size of 1. We first perform 3 warm-up runs, and then run each experiment 10 times and report the mean latency.
> > >
> > > Table 1: Speedup in prefill and decode after 50% structured pruning of the SSM module
> > > | Model | prefill speedup | decode speedup |
> > > | --- | --- | --- |
> > > | Mamba-370m | 1.49x | 1.50x |
> > > | Mamba-790m | 1.50x | 1.47x |
> > > | Mamba-1.4b | 1.50x | 1.48x |
> > > | Mamba-2.8b | 1.98x | 1.95x |
> > >
> > > Using the official optimized kernel of Mamba-1 and the same other configuration, we apply structured pruning to the SSM module, and perform inference with the official accelerated CUDA kernel provided by Mamba. The results below demonstrate that the current structured pruning method can provide effective real-world acceleration for the selective scan process.
> > >
> > > Table 2: Speedup of 50% structured pruning on the SSM module using the optimized `mamba-ssm kernel`
> > > | Model | Dense (ms) | Pruned (ms) | Speedup |
> > > | --- | --- | --- | --- |
> > > | Mamba-370m | 5.530 | 3.210 | 1.72x |
> > > | Mamba-790m | 7.948 | 4.666 | 1.70x |
> > > | Mamba-1.4b | 10.878 | 6.456 | 1.68x |
> > > | Mamba-2.8b | 18.224	| 11.000	| 1.66x |
> > >
> > > In addition, in scenarios where CUDA is unavailable, **speedup from a naive implementation** is also practically meaningful, e.g., on CPU. We further measure the **CPU speedup of the unstructured-pruned model using ONNXRuntime**. We use a sequence length of 400 and a batch size of 1, run each experiment 10 times, and report the mean latency.
> > >
> > > Table 3: CPU Speedup under unstructured pruning of Mamba-370m in ONNXRuntime
> > > | Dense (ms) | 50% sparsity (ms) | Speedup |
> > > | --- | --- | --- |
> > > | 2519 | 1732 | 1.45x |
> > >
> > > Regarding the relatively large cost of the selective scan in our reported inference time, this mainly comes from the naive PyTorch implementation, where the model performs state updates for each new token:
> > >
> > > $$ h_t = \Delta A h_{t-1} + \Delta B x_t $$
> > >
> > > ---
> > >
> > > We would also like to **clarify the scope of this paper**. From an implementation perspective, systematically demonstrating end-to-end acceleration after unstructured pruning, as well as performing complete structured pruning for the full model, still requires substantial additional effort. **This is an important direction for future work.**
> > >
> > > **This is also consistent with common practice in the unstructured pruning literature.** Many influential prior works on unstructured pruning mainly focus on the modeling aspect, because unstructured pruning itself does not naturally translate into speedup. Practical benefits are often a systems-level issue, typically requiring hardware support, and these factors are to a large extent independent of the pruning algorithm itself. Representative examples include:
> > >
> > > - Optimal Brain Damage, NIPS 1989
> > > - Learning both Weights and Connections for Efficient Neural Networks, NIPS 2015
> > > - SNIP: Single-shot Network Pruning based on Connection Sensitivity, ICLR 2019
> > > - PLATON: Pruning Large Transformer Models with Upper Confidence Bound of Weight Importance, ICML 2022
> > > - SparseGPT: Massive Language Models Can be Accurately Pruned in One-Shot, ICML 2023, which explicitly presents only a preliminary study on acceleration and identifies it as an important direction for future work
> > >
> > > ---
> > >
> > > **Finally, we would like to summarize two main points:**
> > >
> > > - Our method is mainly an **unstructured pruning method**, and its primary focus is on modeling, which is consistent with many classical works in the pruning literature. At the same time, the framework can also be extended to structured pruning.
> > > - The additional acceleration experiments we provide already show that our method can still achieve speedup under the optimized mamba-ssm kernel, demonstrating both its practical acceleration ability and its potential for more complete system-level optimization in the future.
> > >
> > > ---
> > >
> > > We sincerely thank you again for your thoughtful feedback. We hope that the additional experiments and discussion above help address your concern.
> > >
> > > Best regards,
> > >
> > > The Authors of Submission 24916

---

### Official Review · Reviewer_k4H7 · 2026-03-02

**Soundness:** 2
**Presentation:** 3
**Significance:** 2
**Originality:** 2
**Overall Recommendation:** 3
**Confidence:** 4

**Summary:**

Addressing the lack of pruning techniques for Mamba-based models, the authors propose SparseSSM. In this framework, the state transition matrix $A$ is pruned using the OBS framework, followed by a component-wise sensitivity analysis for FFN pruning. Compared to existing pruning methods, SparseSSM achieves lower perplexity and higher accuracy across various tasks at equivalent sparsity ratios and demonstrates significant potential for accelerating larger Mamba models.

**Compliance With Llm Reviewing Policy:**

Affirmed.

**Final Justification:**

Despite the rebuttal, some concerns still remain, therefore, I have decided to keep my current score.

**Key Questions For Authors:**

1. Was this tested using the official Mamba-1 CUDA kernels or a standard PyTorch implementation? If using the CUDA kernels, what specific modifications were required to accommodate the pruned matrix $A$? If tested in PyTorch, how would the pruning benefit a practical, hardware-accelerated environment (e.g., Triton or CUDA)?

2. Although perplexity and accuracy is compared across methods in the main table, a direct speed comparison is missing. Could the author provide end-to-end speedup data in the main results (e.g., Table 1) to allow for a better comparison with other methods?

3. Since the primary focus is on pruning $A$, could the author provide a breakdown of the relative impact (in terms of both perplexity and speed) of pruning matrix $A$ versus pruning the FFN components?

**Limitations:**

Yes

**Strengths And Weaknesses:**

### Strengths:
***Soundness***:
1. The paper derives a Hessian matrix estimation and simplifies the computation of OBS pruning specifically tailored for Mamba’s parameter $A$.

2. The authors visualize the Hessian $H_t$ across time steps to validate that a mean Hessian value $H$ can be utilized, providing a solid foundation for pruning the time-invariant matrix $A$.

3. Experimental results show that SparseSSM consistently outperforms other methods at the same sparsity levels across different model sizes.

4. The paper provides a thorough evaluation, including performance metrics, end-to-end speedups, various sparsity patterns/ratios, different aggregation approaches, and testing across Mamba variants (Mamba-2 and MambaVision).


***Presentation***:
1. The paper is well-structured and utilizes high-quality visualizations to effectively clarify complex concepts.

***Significance***:
1. The improvements in perplexity and accuracy over baseline methods are significant. This performance advantage is maintained across different model scales and architectural variations.

***Originality***:
1. The application of the OBS framework to the specific architectural constraints of Mamba is a novel and timely contribution.

### Weaknesses:
***Soundness***:
1. A key contribution of the original Mamba-1 is its specialized CUDA implementation. This paper does not sufficiently address the implementation details; specifically, it remains unclear how the pruning of matrix $A$ functions within the context of highly optimized CUDA kernels.

2. While the paper provides perplexity comparisons across various methods, it lacks a direct end-to-end speedup comparison in the primary results (e.g., Table 1). Without a standardized benchmarking of execution time against other pruning baselines, the core claim regarding the method's efficiency and its ability to accelerate Mamba models remains unsubstantiated.

3. The primary focus of the paper is the pruning of the state transition matrix $A$, yet the evaluation fails to disentangle the relative impact of pruning $A$ versus the FFN components. Without a granular ablation study isolating the effects on both perplexity and speed for each component, it is difficult to assess the actual efficacy and contribution of the proposed $A$-pruning technique compared to standard FFN pruning.

***Significance***:
1. The reported speedup remains somewhat ambiguous. It is unclear if the benchmarks were conducted under practical settings using the Mamba-1 CUDA kernels or a less optimized environment.

***Originality***:
1. While applying OBS to Mamba is a new application, the paper does not introduce particularly surprising or non-trivial algorithmic innovations in the pruning methodology itself.

---

> ### Author Rebuttal · Authors · 2026-03-31
>
> We sincerely thank the reviewer `k4H7` for the careful reading and constructive feedback.
>
> ---
>
> > ***(W1/Q1/Significance):** Unclear how pruning $A$ works with Mamba-1’s specialized CUDA kernels; was this tested with official kernels or PyTorch? What modifications were needed?*
>
> **Reply:** We thank the reviewer for raising this point. Second-order pruning is inherently an unstructured pruning method. We extend our method to the structured pruning setting (details are provided in Sec. 4.3). Since structured pruning reduces the dimensions of the model parameter matrices, it naturally yields speedup across different execution backends, including CUDA, native PyTorch, and CPU implementations.
>
> For speed measurement, we evaluate the pruned model in PyTorch. Specifically, we use `torch.profiler` to measure end-to-end wall-clock speedup, and report the mean and standard deviation over 5 runs. In each run, we perform warm-up forward passes to amortize one-time compilation and caching overheads, and then measure the subsequent forward pass using CUDA events. Our reported speedup does not rely on custom CUDA kernel modifications.
>
> ---
>
> > ***(W2/Q2):** Main results lack a direct end-to-end speed comparison vs other pruning baselines.*
>
> **Reply:** We have added end-to-end speedup results for the other baseline methods under the same evaluation setup, by directly testing the pruned models. The results are summarized below. We will further supplement the full set of experiments in the revision.
>
> We note that, in our speed evaluation, we do not modify the underlying operators and instead directly measure the wall-clock time of a forward pass. As a result, it is difficult to observe practical speedup for unstructured pruning methods such as SparseGPT and MP under this setting. In contrast, when our method is extended to the structured pruning setting, it also achieves better performance.
>
> Table 1: End-to-end speedup of different methods at 50% pruning
> | Model | Method | Dense (ms) | Speedup | Avg. PPL | Avg. zero-shot accuracy |
> | --- | --- | --- | --- | --- | --- |
> | Mamba-370m | Dense | 546.7 | / | 19.05 | 47.49 |
> |  | MP | 545.2 | 1.00 | 310.7 | 42.06 |
> |  | SparseGPT | 545.5 | 1.00 | 3626 | 44.56 |
> |  | SparseSSM (structured) | 329.4 | 1.66 | 23.20 | 46.31 |
>
> ---
>
> > ***(W3/Q3):** Need an ablation separating pruning (A) vs pruning FFN components (perplexity and speed).*
>
> **Reply:** We have added a more detailed ablation study. **For perplexity**, we separately evaluate pruning the SSM module, pruning the FFN, and pruning both jointly. The results are shown below.
>
> Table 2: Results at 30% sparsity by pruning the SSM module, the FFN, or both
> | Model | Method | Wiki. | PTB | C4 | Avg. PPL |
> | --- | --- | --- | --- | --- | --- |
> | Mamba-370m | Dense | 14.32 |  23.46 |  19.37 | 19.05 |
> |  | Pruning SSM module | 14.38 | 23.56 | 19.42 | 19.12 |
> |  | Pruning FFN | 16.22 | 25.94 | 21.65 | 21.27 |
> |  | Pruning both | 16.37 | 26.56 | 21.90 | 21.61 |
>
> **For speed**, to more directly reflect wall-clock acceleration, we further extend our method to structured pruning of the SSM module. Extending structured pruning to the FFN is left for future work. The forward-pass latency of different modules is shown below. This indicates that pruning ssm module brings greater benefits.
>
> Table 3: CUDA forward-pass time of different modules in Mamba-370m
> | **Module** | **inference time (ms)** |
> | --- | --- |
> | selective scan | 535.1 |
> | FFN | 12.03 |
>
> ---
>
> > ***(Originality):** Applying OBS to Mamba may not feel “surprising.”*
>
> **Reply:** We appreciate the reviewer’s perspective, but would like to clarify that our method involves several nontrivial innovations, as summarized below:
> - To the best of our knowledge, we are the first to apply second-order pruning to Mamba-style architectures, explicitly tackling the challenges posed by **time-shared and discretized SSM structures**, which are not addressed by classical OBS-based methods such as SparseGPT or Wanda.
> - Based on our empirical observations of the evolution patterns of SSM states, we further propose a **principled time-aggregation strategy** for mask aggregation, which leads to better practical performance. This component is absent from prior classical OBS-style pruning methods and may also **inspire pruning methods for other models with time-shared parameters**, e.g., RWKV.
> - We also provide practical insights into the heterogeneous pruneability within Mamba blocks, enabling our method to better adapt to the full Mamba architecture.
>
> ---
>
> We thank the reviewer again for these thoughtful and constructive suggestions. We will do our best to address any remaining questions or concerns during the next phase.

---

> > ### Author Rebuttal · Reviewer_k4H7 · 2026-04-02
> >
> > I thank the authors for their detailed rebuttal. However, I remain concerned about the efficiency benchmarking. The authors mentioned that the speed was measured using a naive PyTorch implementation rather than the optimized `mamba-ssm kernel`. In the context of Mamba-1, a naive implementation is not a convincing proxy for real-world performance, as it lacks the key `Hardware-Aware State Expansion` that are essential for its practical viability. Furthermore, the reported inference time, where the `pure selective scan` reportedly costs 45$\times$ more than matrix operations, is highly unusual, given that matrix operations typically account for significantly higher FLOPs. This discrepancy further suggests that the benchmarking was not conducted under a standard, **optimized** setting. Therefore, I prefer to maintain my current score.

---

> > > ### Author Response · Authors · 2026-04-08
> > >
> > > Dear Reviewer `k4H7`,
> > >
> > > We are grateful for your follow-up and for carefully reassessing our efficiency discussion. We would like to further clarify this issue.
> > >
> > > ---
> > >
> > > > *"The reported speedups were measured in a naive, non-optimized setting and therefore do not convincingly reflect real-world efficiency."*
> > >
> > > Our method is mainly an unstructured pruning approach for the Mamba architecture. On top of this, we have already extended our method to structured pruning of the SSM module in the current paper. To further demonstrate the practical speedup **under a standard, optimized setting**, we have now added new acceleration results.
> > >
> > > Using the **official optimized kernel of Mamba-1** and the same configuration, we apply structured pruning to the SSM module, and perform inference with the official accelerated CUDA kernel provided by Mamba. The results below demonstrate that the current structured pruning method can provide effective real-world acceleration for the selective scan process.
> > >
> > > Table 1: Speedup of selective scan under 50% structured pruning on the SSM module using the optimized `mamba-ssm kernel`
> > > | Model | Dense (ms) | Pruned (ms) | Speedup |
> > > | --- | --- | --- | --- |
> > > | Mamba-370m | 5.530 | 3.210 | 1.72x |
> > > | Mamba-790m | 7.948 | 4.666 | 1.70x |
> > > | Mamba-1.4b | 10.878 | 6.456 | 1.68x |
> > > | Mamba-2.8b | 18.224	| 11.000 | 1.66x |
> > >
> > > Following other reviewer's suggestion, **we evaluate the model on a generation task using the official mamba-ssm kernel**, and measure latency with CUDA events, reporting the CUDA **elapsed time** speedup. We use prompt length 2048, decode length 256, batch size 1, perform 3 warm-up runs, then report the mean CUDA elapsed time over 10 runs.
> > >
> > > Table 2: Speedup in prefill and decode after 50% structured pruning of the SSM module
> > > | Model | prefill speedup | decode speedup |
> > > | --- | --- | --- |
> > > | Mamba-370m | 1.49x | 1.50x |
> > > | Mamba-790m | 1.50x | 1.47x |
> > > | Mamba-1.4b | 1.50x | 1.48x |
> > > | Mamba-2.8b | 1.98x | 1.95x |
> > >
> > > In addition, in scenarios where CUDA is unavailable, speedup from a naive implementation is also practically meaningful, e.g., on CPU. Following `SparseGPT`, we further measure the CPU speedup of the unstructured-pruned model using ONNXRuntime, and obtain the following result. We use a sequence length of 400 and a batch size of 1, run each experiment 10 times, and report the mean latency.
> > >
> > > Table 3: CPU Speedup of Mamba-370m under unstructured pruning in ONNXRuntime
> > > | Dense (ms) | 50% sparsity (ms) | Speedup |
> > > | --- | --- | --- |
> > > | 2519 | 1732 | 1.45x |
> > >
> > > Regarding the relatively high cost of the pure selective scan in our reported inference time, this mainly comes from the naive PyTorch implementation, where the model performs a sequential state update for each new token:
> > >
> > > $$ h_t = \Delta A h_{t-1} + \Delta B x_t $$
> > >
> > > ---
> > >
> > > We would also like to **clarify the scope of this paper**. From an implementation perspective, systematically demonstrating end-to-end acceleration after unstructured pruning, as well as performing complete structured pruning for the full model, still requires substantial additional effort. **This is an important direction for future work.**
> > >
> > > **This is also consistent with common practice in the unstructured pruning literature.** Many influential prior works on unstructured pruning mainly focus on the modeling aspect, because unstructured pruning itself does not naturally translate into speedup. Practical benefits are often a systems-level issue, typically requiring hardware support, and these factors are to a large extent independent of the pruning algorithm itself. Representative examples include:
> > >
> > > - Optimal Brain Damage, NIPS 1989
> > > - Learning both Weights and Connections for Efficient Neural Networks, NIPS 2015
> > > - SNIP: Single-shot Network Pruning based on Connection Sensitivity, ICLR 2019
> > > - PLATON: Pruning Large Transformer Models with Upper Confidence Bound of Weight Importance, ICML 2022
> > > - SparseGPT: Massive Language Models Can be Accurately Pruned in One-Shot, ICML 2023, which explicitly presents only a preliminary study on acceleration and identifies it as an important direction for future work
> > >
> > > ---
> > >
> > > **Finally, we would like to summarize two main points:**
> > >
> > > - Our method is mainly an **unstructured pruning method**, and its primary focus is on modeling, which is consistent with many classical works in the pruning literature. At the same time, the framework can also be extended to structured pruning.
> > > - The additional acceleration experiments we provide already show that our method **can still achieve speedup under the optimized mamba-ssm kernel**, demonstrating both its practical acceleration ability and its potential for more complete system-level optimization in the future.
> > >
> > > ---
> > >
> > > We sincerely thank you again for the careful reading and constructive feedback. We hope that the above discussion and the additional acceleration experiments help clarify our position and address your concern.
> > >
> > > Best regards,
> > >
> > > The Authors of Submission 24916

---

### Official Review · Reviewer_VMda · 2026-03-03

**Soundness:** 4
**Presentation:** 3
**Significance:** 4
**Originality:** 3
**Overall Recommendation:** 5
**Confidence:** 4

**Summary:**

The authors introduce an adaptation of the OBS pruning framework for state-space models. They provide an overview of how SSMs are different to traditional transformers, present a derivation of the OBS saliency metrics for SSMs. and an algorithm for pruning SSMs. The method outperforms existing pruning methods, both ones that are specialized to SSMs (MambaShedder) and ones that are not (SparseGPT, WANDA). Overall I am positive on the paper.

**Compliance With Llm Reviewing Policy:**

Affirmed.

**Key Questions For Authors:**

-- Could the authors please explain exactly how they measured the speed-ups in Table 2. To my knowledge, sparsity does not typically provide speed-ups in e.g. PyTorch except for at very high sparsity levels, without specialized kernels. Why do the authors believe speed-ups were achievable here?

-- I believe the algorithm could use more explanation. Can you add a dedicated section where you explain the steps of the algorithm? I believe some intuition is explained in the text, but this algorithm is far less intuitive than SparseGPT or OBS, and could use elaboration.

-- I would appreciate if the authors could clean up the theorems/derivations to an extent. Every step should follow logically and all results, where borrowed from OBS, should be explicitly stated. Otherwise, it's hand-wavey and hard to verify.

**Limitations:**

yes

**Strengths And Weaknesses:**

Strengths:

-- I am positive on the paper overall. It provides a nice adaptation of existing pruning methods and frameworks (OBS/SparseGPT) to state-space models such as Mamba.

-- The derivations generally seem correct.

-- The explainer figures on the unique Hessian structure of SSMs e.g. Figure 2 are quite informative and interesting.

-- The method outperforms the existing pruning baselines for this task by a significant margin.

Weaknesses:

-- As a minor comment, the authors have "training-free" bolded in the opening section which introduces one-shot pruning. In my mind one-shot should be bolded since presumably this refers to introducing a concept, and I believe one-shot pruning is the dominant terminology that is used, not training-free. I understand this is to be contrasted with training-based methods, but I think it would be clearer and more standard to say one-shot.

-- In section 3.1, the notation $B^{\prime}$ is not introduced from what I can tell. Did you mean $\hat{B}$?

-- I believe the presentation of Theorem 3.1 needs to be improved. For example you use boldface L and regular L, when these refer to the same object. Moreover, as a more general comment, I find the theorem to be a bit cryptic. For ex, the sentence "Because $w$ acts multiplicatively and component-wise on $h_{b,t−1,d,n}$, cross-partials across different $(b, t, d, n)$ are negligible. The Hessian is effectively
block-diagonal in these coordinates." Please just write it in math or it's harder to verify.

-- The authors use $p$ to refer to the power-law decay parameter and $p$ to refer to the sparsity budget. I think these are different quantities, right? If so, please correct the notation.

---

> ### Author Rebuttal · Authors · 2026-03-31
>
> We sincerely thank the reviewer `VMda` for the careful reading, positive assessment, and detailed suggestions on presentation and verification. Below we respond point‑by‑point.
>
> ---
>
> > ***(W1):** “training-free” is bolded; “one-shot” is the dominant terminology.*
>
> **Reply:** We agree and will revise the introduction to emphasize one-shot pruning as the primary term, using “training-free” only as a clarifier in contrast to fine-tuning-based pruning pipelines.
>
> ---
>
> > ***(W2):** Notation in Sec. 3.1 seems undefined.*
>
> **Reply:** This is a presentation issue that we will fix. Here, $ B' $ was intended to denote the pre-discretization SSM input parameter, while we used the prime only to distinguish it from the symbol $ B $ that is already reserved in the paper for the batch size.
>
> By contrast, $\hat{B}$ denotes the discretized parameter actually used in the recurrence after the discretization step. In the revision, we will explicitly define both $B'$ and $\hat{B}$ at their first appearance.
>
> ---
>
> > ***(W3):** Theorem 3.1 is cryptic; the “block diagonal” claim is described in prose and is hard to verify.*
>
> **Reply:** We will revise the presentation of Theorem 3.1 as follows:
> - We will adopt a unified notation for $L$, using $\mathcal{L}$ to denote the loss function, and we will clearly distinguish it from the sequence length $L$.
> - We will use mathematical expressions as much as possible, and revise the cited sentence as follows:
>
> Consider a scalar coordinate $t$, we have
> \begin{equation}
> w = (\Delta A)_{b,t,d,n},
> \end{equation}
>
> At a fixed time step $t$, $h$ satisfies
>
> $$
> \Delta A_t \in \mathbb{R}^{B \times D \times N} \, h_{t} \in \mathbb{R}^{B  \times D \times N}
> $$
>
> Since the computation takes the form
>
> $$
> h_{b,t,d,n} = w \odot h_{b,t-1,d,n} + (\Delta B_{b,t} u_{b,t})_{d,n}.
> $$
>
> Here, $u_{b,t}$ denotes the input at step $t$, which is fixed with respect to $w$. $\Delta B_{b,t}$ depends on the input, but does not depend on the scalar variable $w$. By direct differentiation of the local update,
>
> $$
> \frac{\partial h_{b,t,d,n}}{\partial w} = h_{b,t-1,d,n}.
> $$
>
> The subsequent proof is consistent with Eq. 16 in Appendix A.
>
> ---
>
> > ***(W4):** The symbol $p$ is used for both the power-law exponent and sparsity budget.*
>
> **Reply:** Thank you for pointing this out. We will revise the notation here by defining $p$ as the power-law parameter and $μ$ as the sparsity budget.
>
> ---
>
> > ***(Q1):** How were speedups measured? Why would sparsity give speedups in PyTorch without specialized kernels?*
>
> **Reply:** Our method is originally an unstructured pruning approach. To directly observe end-to-end speedup, we further extend it to the structured pruning setting, as described in Section 4.3. Specifically, the reported end-to-end speedups **come from structured pruning with dimension reduction** on the SSM state channels: we prune along the second axis of $ x_{proj} $ and resize the output dimension of $ A_{log} $ accordingly. Under structured pruning, the reduced matrix dimensions naturally lead to noticeable acceleration even without any specialized kernels.
>
> For our speed measurement, we use `torch.profiler` to measure end-to-end wall-clock speedup, and report the mean and standard deviation over 5 runs.
>
> ---
>
> > ***(Q2):** Can you add a dedicated section where you explain the steps of the algorithm?*
>
> **Reply:** We will add a concise “Algorithm Overview” subsection in Section 3.3 to explain
> - what statistics $(S_t=\mathbb{E}[h_t^2])$ are accumulated and how they are accumulated (line 6 in Algorithm 1),
> - how the computation is aggregated across the batch dimension (line 10),
> - how the power-law weighting emphasizes informative prefix timesteps. (line 12), and
> - how the indices of the final parameters to be pruned are selected (line 14).
>
> ---
>
> > ***(Q3):** Cleaning up the theorems / derivations.*
>
> **Reply:** We agree and will clean up the theorems and derivations accordingly. In Appendix A, we already state the diagonal OBS saliency form and then derive the SSM-specific Hessian estimate. We will rewrite this section to explain
> - Our use of OBS is primarily in the saliency computation (Eq. 9), and in the approximation that ignores first-order information (Eq. 14). Beyond these parts, the remaining derivations are developed step by step according to the underlying logic and the structure of the model.
> - We will further make the currently ambiguous textual explanations clearer, including the issue raised in **(W3)**, and we will also revise the notation to ensure consistency throughout.
>
> ---
>
> To conclude, we would like to sincerely thank the reviewer again for your careful reading, constructive criticism, and concrete suggestions.

---

> > ### Author Rebuttal · Reviewer_VMda · 2026-04-03
> >
> > Dear Authors,
> >
> > Thank you for your detailed responses. As I said, I am positive on the paper and find it a good contribution.
> >
> > I find the presentation to be problematic in the initial version. The fixes you propose to my comments are satisfactory. However, I would ask the authors to carefully re-check that there is no duplicate notation (I found several, but perhaps not all) and that each step in your theorems is clear and rigorous (right now the theorems are a bit too hand-wavey). Provided the presentation issues I brought up are fixed, the paper is a clear accept in my view. I understand you cannot edit the paper now, but I am trusting you will do so for the camera-ready version if accepted. As a result I am going to increase my score by one point.

---

> > > ### Author Response · Authors · 2026-04-08
> > >
> > > Dear Reviewer `VMda`,
> > >
> > > Thank you very much for revisiting our work and for deciding to raise the score. We truly appreciate your **finding our paper a good contribution**, which is very encouraging to us.
> > >
> > > We are especially grateful for the presentation issues you pointed out in our paper. Your suggestions have been very helpful in improving the paper. We will carefully double-check (i) the notation throughout the paper, (ii) the derivation in Theorem 1 and its proof, and (iii) any other places where the presentation may still be unclear, and we are committed to comprehensively **fixing these issues in the camera-ready version if the paper is accepted**. We are also grateful for the time and care you devoted to reading our rebuttal.
> > >
> > > Best regards,
> > >
> > > The Authors of Submission 24916

---

### Decision · Program_Chairs · 2026-04-30

**Decision:**

Accept (regular)

**Comment:**

This paper proposes SparseSSM, a training-free OBS-style pruning framework for Mamba-based SSM models. The paper is a resubmission and has benefited from several rounds of revision. In its current form, it has improved considerably, and the reviewer sentiment is positive overall. The authors have effectively addressed some remaining concerns. I believe this work makes a valuable contribution and should be accepted.